# Performance of automated methods for flash flood inundation mapping: a comparison of a DTM filling and two hydrodynamic methods

Nabil Hocini[1,*], Olivier Payrastre[1,*], François Bourgin[1,2], Eric Gaume[1], Philippe Davy[3], Dimitri Lague[3], Lea Poinsignon[4], and Frederic Pons[4]

[1]GERS-LEE, Univ Gustave Eiffel, IFSTTAR, F-44344 Bouguenais, France
[2]Université Paris-Saclay, INRAE, UR HYCAR, 92160 Antony, France
[3]Géosciences Rennes, 35042 Rennes, France
[4]Cerema Méditerranée, 13290 Aix-en-Provence, France
[*]These authors contributed equally to this work.

**Correspondence:** Nabil HOCINI (nabil.hocini@univ-eiffel.fr), Olivier PAYRASTRE (olivier.payrastre@univ-eiffel.fr)

**Abstract.** Flash floods observed in headwater catchments often cause catastrophic material and human damage worldwide. Considering the large number of small watercourses possibly affected, the use of automated methods for flood inundation mapping at a regional scale can be of great help for the identification of threatened areas and the prediction of potential impacts of these floods. An application of three mapping methods of increasing level of complexity is presented herein, including a Digital Terrain Model (DTM) filling approach (Height Above Nearest Drainage/Manning Strickler or HAND/MS) and two hydrodynamic methods (caRtino 1D and Floodos 2D). These methods are used to estimate the flooded areas of three major flash floods observed during the last ten years in South-Eastern France: the 15th of June 2010 flood on the Argens river and its tributaries (585 km of river reaches), the 3rd of October 2015 flood on small coastal rivers of the French Riviera (131 km of river reaches) and the 15th of October 2018 floods on the Aude river and its tributaries (561 km of river reaches). The common features of the three mapping approaches are their high level of automation, their application based on a high-resolution (5m) DTM, and their reasonable computation times. Hydraulic simulations are run in steady-state regime, based on peak discharges estimated using a rainfall-runoff model preliminary adjusted for each event. The simulation results are compared with the reported flood extent maps and the high water level marks. A clear grading of the tested methods is revealed, illustrating some limits of the HAND/MS approach and an overall better performance of hydraulic models solving the shallow water equations. With these methods, a good retrieval of the inundated areas is illustrated by Critical Success Index (CSI) median values close to 80%, and the errors on water levels remain mostly below 80 cm for the 2D Floodos approach. The most important remaining errors are related to limits of the DTM such as the lack of bathymetric information, uncertainties on embankment elevation and to possible bridge blockages not accounted for in the models.

# 1 Introduction

Flash floods represent a significant part of flood related damages worldwide, particularly in regions prone to large rainfall accumulations over limited duration - typically several hundreds of mm in a few hours. For instance, in France eight floods caused insurance losses exceeding 500 million euros over the period 1989-2018, among which 4 were flash floods (CCR, 2020). Therefore, the development of efficient risk management policies for small upstream watercourses has become a particularly important issue. However, the capacity to face flash-flood related risks is still highly limited by the very large number of small rivers, and by the specific features of flash floods: high unit discharge values, fast evolution in time, high spatial heterogeneity, and low predictability.

One crucial aspect to mitigate the flash floods risks is to improve the flood hazard mapping on small watercourses, typically with upstream drainage areas starting at a few km$^2$. Such information is essential for an appropriate development of prevention policies and crisis management plans. If available, it may be particularly helpful for stakeholders to: (i) facilitate risk identification and awareness, (ii) speed-up decision-making before and during the crisis.

The development of detailed flash flood hazard mapping still suffers from serious limitations due to the lack of descriptive data for small rivers (topography, bathymetry,..) and their flood regimes. However, a large increase in resolution and accuracy of Digital Terrain Models (DTMs) has been observed in the last few years, particularly with the development of Lidar, and DTMs with a resolution of less than 10 m are now widely available even if their accuracy remains heterogeneous (Schumann and Bates, 2018). This evolution makes it possible to run hydraulic simulations on small rivers (Lamichhane and Sharma, 2018). Even if information on bathymetry is still rarely available, recent advances have been achieved in estimating unknown bathymetry or river channel geometry based on remote sensing or local at-site surveyed data (Gleason and Smith, 2014; Neal et al., 2015; Grimaldi et al., 2018; Brêda et al., 2019). Regionalized hydrological approaches also progressively help improve knowledge on flood regimes of upstream watercourses (Aubert et al., 2014).

The context is therefore increasingly favorable for the development of flood hazard mapping on small rivers prone to flash floods. However, this requires efficient mapping methods which can be easily applied on detailed river networks at regional scales. For instance, in France the entire stream network includes 120.000 km of rivers of more than 1 meter width, whereas flood hazard information is concentrated on the 23.000 km of main rivers, corresponding to the network covered by the Vigi-crues national flood forecasting service. It can thus be estimated that about 100.000 km of small rivers should be documented with hazard information to ensure a comprehensive coverage. Hence, there is a need for automated and fast computing methods, which excludes both the mobilization of hydraulician's expert knowledge and thorough calibration of models. An appropriate representation of uncertainties requiring to run a diversity of scenarios with different boundary conditions and/or parameters, and/or the integration of mapping approaches in real-time forecasting chains, may make the question of computation times even more critical (Savage et al., 2016; Dottori et al., 2017; Morsy et al., 2018).

Several flood inundation mapping methods which meet the objective of a high level of automation have gradually been developed in the recent years. These methods can be classified into two main categories: i) hydraulic approaches solving the

Saint-Venant shallow water equations (SWE) in 1 or 2 dimensions, ii) direct DTM filling approaches based on preliminary retrieving/estimation of the local discharge/water height relation.

Hydraulic 2D SWE models have been applied for a long time at regional and continental scales (Pappenberger et al., 2012; Alfieri et al., 2014; Sampson et al., 2015; Dottori et al., 2016; Schumann et al., 2016), but at resolutions (100 m to 1 km) incompatible with the representation of small rivers (Fleischmann et al., 2019). The main factors limiting the resolution were both the availability of global high-resolution DTMs (Schumann and Bates, 2018), but also the computation capacities. For instance, Savage et al. (2016) consider that resolutions finer that 50 m offer a limited gain due to other sources of uncertainties, while resulting in a large increase of computational expense; Teng et al. (2017) confirm that 2D hydrodynamic models have for a long time been unsustainable for areas larger than 1000 km$^2$ at resolutions of less than 10 m. However, the progress in efficiency of algorithms and in parallel computation now enable regional to continental applications at a 30 m resolution (Morsy et al., 2018; Wing et al., 2017, 2019). Several examples of flood mapping applications at finer resolutions (<10 meters) have also been recently presented, based on a large variety of models: DHD-Iber (Cea and Bladé, 2015), Floodos (Davy et al., 2017), Lisflood FP (Neal et al., 2018), PRIMo (Sanders and Schubert, 2019), SRM (Xia et al., 2017). Specific applications to flash floods have been proposed using Iber (García-Feal et al., 2018), BreZo (Nguyen et al., 2016), and B-flood (Kirstetter et al., 2021). Finally, in addition to high resolution 2D models, 1D SWE models may also be applied based on cross-sections extracted from high-resolution DTMs (Choi and Mantilla, 2015; Pons et al., 2014; Le Bihan et al., 2017; Lamichhane and Sharma, 2018), also showing interesting results in terms of accuracy and offering lower computation times.

Direct DTM filling approaches have been developed more recently. All these methods are based on a local discharge/water height relationship determined from i) the cross-section and longitudinal profile geometries, and ii) a local hydraulic formula: Manning-Strickler (Zheng Xing et al., 2018; Zheng et al., 2018; Johnson et al., 2019; Garousi-Nejad et al., 2019) or Debord (Rebolho et al., 2018). The cross-section geometry is either extracted locally from the DTM for the AutoRoute method (Follum et al., 2017, 2020), or averaged at the river reach scale based on a Height Above Nearest Drainage (HAND) raster (Nobre et al., 2011) for the following methods : f2HAND (Speckhann et al., 2017); Geoflood (Zheng et al., 2018); MHYST (Rebolho et al., 2018); Hydrogeomorphic FHM (Tavares da Costa et al., 2019). These approaches are very efficient in terms of computation times, and can therefore be suitable for real time inundation forecasting at continental scales (Liu Yan Y. et al., 2018), or for probabilistic or multi-scenario modelling (Teng et al., 2017). However, because of their high level of simplification, these approaches may not reach the same level of accuracy as SWE 2D approaches (Afshari et al., 2018; Wing et al., 2019).

This paper proposes a new contribution to the question of flood hazard mapping, focused here on the specific context of flash floods observed in small headwater catchments. The main question addressed is the following: which performance in inundation mapping can be achieved for the small to intermediate rivers prone to flash floods (from 5 km$^2$ to 2000 km$^2$ catchment surface)? The use of automated approaches based on very high resolution DTM (typically 5 meters or less) is considered here as a necessity considering the limited width of rivers to be covered, and their very large number at a regional scale. A simplification of mapping approaches can be considered as an advantage to limit the computation times and facilitate the application at regional scales, while another objective is to remain as close as possible to an expert modelling in terms of accuracy.

Three approaches of increasing level of complexity are compared here: DTM filling (HAND/Manning Strickler), 1D SWE (caRtino 1D), and 2D SWE (Floodos). The tested Floodos 2D model remains simplified if compared to more conventional 2D SWE approaches (steady-state computation on the DTM mesh, inertial terms neglected in SWE). A comprehensive evaluation and validation exercise is proposed based on varied case studies, corresponding to three recently observed major floods in South-Eastern France. The three selected case studies are particularly well documented in terms of observation and validation data (peak discharges, observed inundation extent, high water level marks). The mapping methods are evaluated based on their ability to reproduce both the actual inundation extents and the high water levels.

The paper is organized as follows: section 2 presents the various tested mapping methods ; the evaluation approach and the selected case studies are presented in section 3 ; the results are presented in section 4 and discussed in section 5 ; section 6 summaries the main conclusions of this work.

## 2 Description of selected flood mapping approaches

### 2.1 Height Above Nearest Drainage/Manning-Strickler (HAND/MS) approach

Rennó et al. (2008) and Nobre et al. (2011) have introduced the height above nearest drainage concept, which is a terrain descriptor representing the height of each DTM grid cell in reference to the nearest stream cell along the drainage path. Nobre et al. (2016) first suggested to use HAND contours for flood hazard mapping. The approach has been recently improved by Zheng Xing et al. (2018) and Zheng et al. (2018), who proposed the GeoFlood method enabling flood mapping based on any input discharge value, by deriving a local height/discharge relation.

The HAND/MS approach applied here is similar to the GeoFlood method. A HAND raster derived from the DTM is used to estimate the average geometry of the river channel for each river reach, namely the evolution of wetted perimeter and wetted area as a function of water height. This information is then used to estimate a local rating curve (discharge/water height relation) based on the Manning-Strickler formula. Any river discharge can then easily be converted into a mean water height in the considered river reach, and into the corresponding inundation extent by comparison with the HAND raster values (Fig. 1). All this computation workflow was implemented here based on TAUDEM libraries (https://hydrology.usu.edu/taudem). The main difference with the GeoFlood approach lies in the delineation of the stream network. A conventionnal approach based on D$\infty$ flow directions (Tarboton, 1997) is used here instead of the Geonet approach used in GeoFlood (Zheng et al., 2018). Possible problems in the determination of the stream network are solved by a pre-treatment of the DTM to eliminate remaining obstacles such as bridges.

This HAND/MS approach is very fast in terms of computation times and has already been applied at a continental scale on very high resolution DTMs (Liu Yan Y. et al., 2018). However it is based on several important assumptions. First, the cross-sectional geometry and water level are averaged and supposed to be uniform for each river reach. Therefore, backwater effects due to longitudinal slope and cross section shape variations along one river reach, and/or due to downstream limit conditions, are not represented. Second, longitudinal discharge variations along each river reach cannot be accounted for. Third, the inundation depth at each point of the floodplain depends only on its relative elevation above its nearest downstream

drainage point (i.e. the HAND raster value), independently of the real hydraulic connections. This may result in discontinuities : neighbour pixels having similar elevations may be related to different drainage points and hence be attributed different hand values. This is particularly true in the case of flat and wide floodplains and at confluences where neighbour pixels may be connected to different river reaches. In this latter case, the water levels considered for the inundation mapping will also be different for the two neighbour points. Johnson et al. (2019) conclude that significant errors may be observed for both low-

order upstream river reaches, and downstream and flat floodplains.

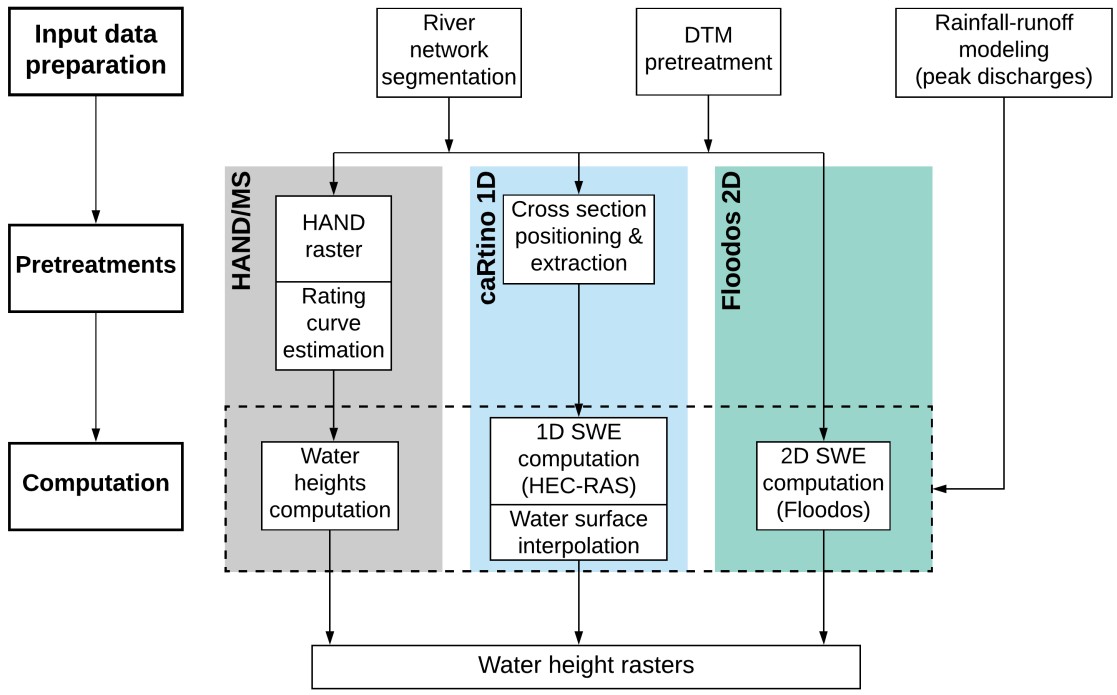

**Figure 1.** Synthetic representation of the simulation workflow for the 3 selected flood mapping methods

## 2.2   caRtino 1D approach

The caRtino 1D method has initially been proposed by Pons et al. (2014). Based on the DTM, it automatically builds the input files for some standard 1D SWE hydraulic models such as HEC-RAS (Brunner, 2016; Brunner et al., 2018) or Mascaret (Goutal et al., 2012). HEC-RAS 5.0.4 has been used herein. The structure of the hydraulic model is defined by an automatic positioning

of cross-sections at selected distances along the river network, and then extracting the cross-sectional profiles from the DTM. Since the distances between cross-sections may have a significant impact on 1D hydraulic simulation results (Ali et al., 2014), the cross-sections are positioned with the double objective to limit their spacing and avoid overlapping. This is achieved in the following way: i) a constant distance between cross-sections is first used (50 meters in this application) ; ii) a first hydraulic run is conducted to estimate the width of the floodplain ; iii) the distance between cross-sections is then set to a proportion

of the floodplain width (here 30%), and the cross-sections are reoriented if crossing each other. Although the positions of cross-sections may be modified manually to improve the accuracy of local studies, this possibility was not considered here. A post-treatment of the simulated water longitudinal profiles enables to retrieve the water surface elevations and the water heights on the grid of the DTM. This method has already been evaluated for flash flood forecasting purposes, showing an interesting capacity to represent the observed inundations (Le Bihan et al., 2017). The caRtino 1D version used here corresponds to an evolution (reprogrammation in R) of the initial software.

Since this approach enables a full resolution of SWE equations, in steady state in the presented applications, it accounts for backwater effects and longitudinal channel geometry variations within river reaches. Its main limits, already identified by Le Bihan et al. (2017), lie in the 1D scheme which may not be adapted in areas with complex hydraulic features. The automated application may also be a source of significant errors. Cross-sections may not be positioned perpendicular to the stream main axis in meandering rivers, leading to cross-section shape distortions. Cross-sections may also be truncated leading to ignore locally part of the floodplain in the computations. Headwater losses due to hydraulic singularities such as bridges cannot be easily integrated. No distinction is made between the river bed and the floodplain and the floodplain continuity between successive cross-sections is not embedded in the model. These limits may have a particular importance in areas with very wide floodplains, perched river beds, or at river confluences.

## 2.3 Floodos approach

Floodos is a 2D SWE computation code developed by Davy et al. (2017). It represents the hydrodynamic module of the Eros program, aiming at simulating erosion processes. The SWE resolution method is running directly on the DTM grid, and is based on a particle-based so-called "precipiton" approach, which consists in propagating elementary water volumes on the water surface. The inertial terms are neglected in the SWE resolution scheme, which may result in errors in case of sudden changes in flow direction and in the vicinity of obstacles. However, the method enables a fast computation of the stationary solution thanks to the choice of a judicious numerical scheme. Davy et al. (2017) indicate the CPU time changes approximately linearly with the number of pixels of the computation domain. They compared Floodos with the widely used 2D LISFLOOD-FP model (Bates et al., 2010). They obtained similar results and faster computation times with Floodos, although they mention this comparison should not be considered as a benchmark..

The Floodos model requires a careful verification of the convergence, since the choice of a too large precipiton volume may result in a bad convergence and significant errors (overestimation of water levels). The convergence verification has been automated here by using a new version of Floodos, enabling to reduce progressively the precipiton volume during the computation. Three decreasing precipiton volumes, defined in accordance with the criterion proposed by Davy et al. (2017), have been systematically applied within each run to ensure the convergence to the right solution.

 **3 Evaluation approach**

## 3.1 Principle of evaluation based on observed flood events

The capacities of the three mapping approaches to reproduce actually observed inundation patterns (i.e. inundated areas and high water level marks) are compared.

The main advantage of this approach is that actual observations are used as reference, when reference expert simulations, with their limits and uncertainties, have often been used in previous similar studies. Hence, a total uncertainty is measured here, including all uncertainties sources in the input data (DTM, but also actual discharge values that are only inaccurately known), parameters (roughness values), and simulation methods.

A possible drawback is that the uncertainties in the input data, particularly in the estimated discharges, may be relatively large and may dominate other sources of uncertainties associated with the simulation methods. For this reason, well documented case studies have been selected here, both in terms of peak discharges of the flood and observed inundation patterns. Particularly, extensive sets of peak discharge estimates on ungauged river sections, gathered within the HyMeX program (Ducrocq et al., 2019), are available for each selected event. These largely complement flood discharge data that are available for a generally limited number of stream gauging stations.

Additionally, a comprehensive knowledge of the inundation characteristics is available for the selected flood events, thanks to a high number of high water marks (HWM) and to field observations of the limits of the inundated areas. The HWM data was extracted from the french national HWM database (https://www.reperesdecrues.developpement-durable.gouv.fr). This data is systematically checked before incorporation in the database and therefore should not include large errors. However, errors up to 50 cm should be considered as common considering the accuracy of topographic surveys (HMW location and elevation), and/or possible inappropriate choice of HWMs locations (increase of water surface elevation in front of obstacles, capillary rise of moisture in walls, ..). Some larger errors may also remain for a very limited number of HWMs, and may result locally in large estimated simulation errors. But all these error sources are common to the 3 methods and should not affect the comparison results. The detailed mapping of inundation extents, available for the Argens 2010 and Aude 2018 events, was achieved by local authorities based on field surveys in the weeks following the floods. This data should have a good accuracy even if it may have been locally interpolated between field observation points.

## 3.2 Flood events selected

The rivers of Southern France are known to be prone to flash floods. This region has experienced a large number of catastrophic flash flood events in the past, including the three floods selected in this study, which are presented on Fig. 2.

The first selected flood occurred in the Argens river watershed (2750 $km^2$) on the $15^{th}$ of June 2010. It is certainly one of the most catastrophic event observed in the last decades in this region: twenty-five victims and 450 million euros of insured losses were reported (source Caisse Centrale de Réassurance - CCR, vehicles excluded). The flood particularly affected the eastern part of the Argens catchment area, where the maximum accumulated rainfall locally exceeded 400 mm in 36 hours. Peak discharges were estimated at about 450 $m^3.s^{-1}$ on the Nartuby tributary river (222 $km^2$), 480 $m^3.s^{-1}$ on the Florieye

river (89 $km^2$), and 2500 $m^3.s^{-1}$ on the downstream part of the Argens river (Payrastre et al., 2019). The length of the river network selected for the hydraulic simulations is 585 km. 557 high water marks are available for the evaluation, as well as the observed limits of inundated areas.

The second event occurred on the $3^{rd}$ of October 2015 and hit several small rivers of the Alpes Maritimes coastline (French Riviera). A storm cell formed at the eastern edge of the Var river and ran along the coastline with a stationary regeneration lasting two hours. A maximum accumulated rainfall of 220 mm in 24 hours (150 mm in 2 hours) was observed in a 30km by 15km band along the coastline. The main rivers such as the Var and Loup rivers were hit only in their downstream part and had limited reactions, but major floods were observed on small coastal rivers such as the Brague river (66 $km^2$, peak discharge >400 $m^3.s^{-1}$), the Riou de l'Argentière river (48 $km^2$, >300 $m^3.s^{-1}$ ) and the Grande Frayère river (22 $km^2$, >180 $m^3.s^{-1}$). These floods caused the death of 20 people and considerable material damage, insured losses being estimated at 520 million euros for this event (CCR, vehicles excluded). The river network selected for the hydraulic simulations is 131 km in length. 428 high water marks have been used for the evaluation.

The last event occurred on the $15^{th}$ of October 2018 in the intermediate part of Aude river watershed (5050 $km^2$), where a accumulated rainfall of more than 300 mm in 24 hours was locally recorded (Caumont et al., 2020). Several tributaries of the Aude river had very strong flood reactions: Lauquet river (196 $km^2$, peak discharge of about 880 $m^3.s^{-1}$), Trapel river (55 $km^2$, >300 $m^3.s^{-1}$), and Orbiel river (253 $km^2$, 490 $m^3.s^{-1}$). These tributaries caused a large flood of the Aude main river immediately downstream the town of Carcassonne. Numerous villages were heavily flooded and suffered large damages. 14 fatalities were reported, several bridges and roads were destroyed, and the insured losses exceeded 200 million euros (CCR, estimation still to be consolidated). The hydraulic simulations were performed on a 569 km river network. 1082 high water marks and the observed limits of inundated areas have been used for the evaluation.

### 3.3 Common input data and simulation workflow

The main steps of the simulation workflow are presented on Fig. 1. The mapping approaches are all implemented on segments of the computation domains (river reaches), the segmentation being based on the structure of the hydrographic networks (confluences). A 5 $km^2$ upstream catchment surface has been selected as a lower limit to define the river network (1 $km^2$ for the Alpes Maritimes case study). One independent computation is conducted on each river reach. This principle of segmentation of the computation domains facilitates the computation over large areas (several hundreds km of rivers in the case studies presented), and has the advantage to easily enable parallel computing if necessary (not implemented here). For SWE hydraulic approaches, the computation is extended 1 km downstream each river reach to limit the influence of the downstream boundary condition on the results (normal flow depth). The results are then combined, taking the maximum water height in areas where the results of several sections overlap - typically areas downstream confluences. For the HAND/MS approach, to avoid merging significantly different channel geometries, the river reaches have been subdivided to limit their length to a maximum of 1500 m, as recommended by Zheng et al. (2018).

The simulations are all run in steady state regime based on estimated flood peak discharges for each river reach. The steady state assumption may lead to an overestimation of the inundation extent and depths if the volume of the flood wave is limited

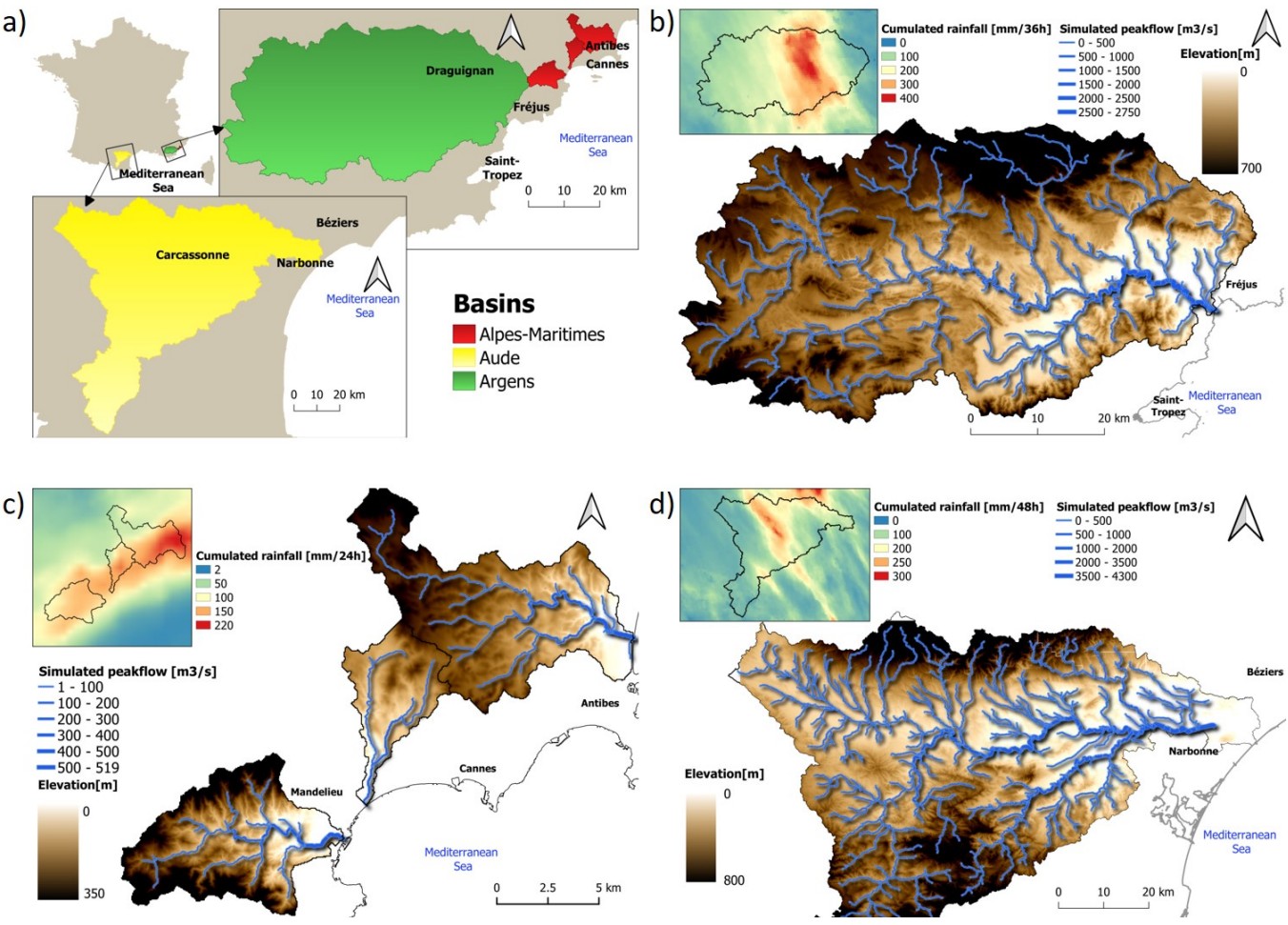

**Figure 2.** Presentation of the 3 considered areas and flood events: a) location of watersheds, b) Argens 2010, c) Alpes Maritimes 2015, d) Aude 2018 (Map data : © IGN, © Météo France)

in comparison with the storage capacity of the floodplain. This assumption is considered here as reasonable since the widths of the floodplains do not exceed several hundred meters, and therefore the corresponding floodplain storage capacities should remain limited. The computation based on flood peak discharges may also lead to an overestimation of backwater effects at

confluences, because of the underlying assumption that maximum peak discharges occur simultaneously for all river branches at a confluence. Lastly, the variations of peak discharges along each river reach are not represented, but these variations are limited since the delineated river reaches have a limited length.

To enable the comparison between mapping methods, the simulations are run using strictly identical inputs: DTMs, peak discharges, and friction coefficients.

The DTMs used are extracted from the IGN RGE Alti ® product and have all a 5 m resolution. In the areas selected as case studies, they are mainly derived from Lidar data (20 cm mean elevation accuracy). However some parts of the areas are still covered with photogrammetry data (70 cm mean elevation accuracy). Bathymetric surveys are not available in the considered areas. The lidar campaigns are conducted in low flow periods, but in some places, the permanent water surface is captured in the DTM. Fortunately, the low flow discharges in the small Mediterranean rivers considered here are limited - some being

ephemeral streams - and the existing DTM generally provide acceptable estimates of the river cross-sections. As the methods may be sensitive to some categories of errors present in DTMs, including for instance the presence of bridges or other structures crossing the rivers and not cleaned up, an automatic pre-treatment has been systematically applied to eliminate any remaining bridges in the river beds.

        The peak discharges are estimated based on preliminary rainfall-runoff simulations obtained with the Cinecar distributed

model (Naulin et al., 2013). The Antilope J+1 rainfall product of Météo-France (Champeaux et al., 2009), combining radar and point rainfall records, was used as input data. The model was calibrated for each event against available discharge observations to limit as far as possible the errors on peak discharges used as input of hydraulic simulations. Overall, the differences between simulated and observed peak discharge do not exceed +-20% (see Fig. 3). However, observations are mainly based on post-flood surveys and may have large uncertainties, as indicated by error bars on Fig. 3. Moreover, observations are not available

at each branch of the considered river networks. Therefore, the simulated peak discharges obtained from the rainfall-runoff model may locally differ significantly from the actual ones..

        The same Manning's roughness coefficients are used in all computations and for all river reaches. They are fixed to n=0.066, which can be considered as a reasonable value for flash floods according to the analysis of available post event surveys data in the considered case studies (Lumbroso and Gaume, 2012). Lower roughness values were also tested, but resulted in negative

bias on water levels for the three case studies and the three methods.

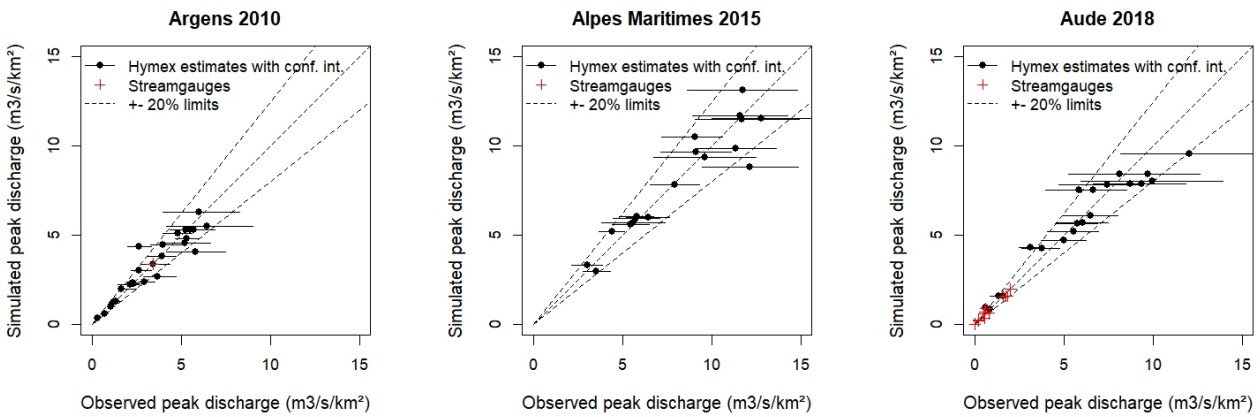

**Figure 3.** Observed versus simulated peak discharges with the Cinecar rainfall-runoff model for the three simulated flood events

### 3.4 Evaluation criteria

#### 3.4.1 Comparison of simulated vs. observed or reference flood extents

The results are evaluated here by comparing the simulated flood extent and the observed one. Overlapping these two areas enables to distinguish four zones (see Fig. 4): the hit zone (a) including areas flooded in both simulation and observation; the

false alarm zone (b), corresponding to areas flooded only according to simulation results ; the miss zone (c), which is included only in the observed flooded area ; and the dry zone (d), corresponding to areas located outside the inundation extent for both simulation and observation. The respective areas of zones (a), (b), and (c) are finally synthesized in the form of a Critical Success Index (CSI), computed for each river reach:

$$CSI = \frac{a}{a+b+c} \tag{1}$$

CSI values range from 0% (no common area between simulation and observation) to 100% (perfect match). Since this metric cumulates overestimation (b) and underestimation (c), it may decrease significantly even for simulation results which appear visually to fit well the observations. Fleishmann et al. (2019) consider that hydrodynamic models with CSI scores greater than 65% at reach scale show satisfactory results.

A possible drawback of this metric is that observations of actual flood extents are generally gathered for major floods events,

with the objective to establish historical references as support of flood risk management policies. These flood events are likely to be valley-filling, which is clearly the case for the three events considered here. This makes the retrieval of the flood extent much easier to achieve with modeling tools, and may mask the differences of performance between the different competing approaches.

#### 3.4.2 Comparison of water surface elevation with high water marks data

The elevations of the simulated water surface and of available high water marks are compared here (see Fig. 4). This results in several hundreds of point differences between simulated and observed water levels. Negative values indicate an underestimation of water levels by the model, while positive values indicate an overestimation. If the model does not predict any inundation at the position of the high water mark, it is considered that the predicted water height is 0 m, and thus the computed error corresponds to the elevation of the high water mark above ground.

In situations where the geomorphologic floodplain is entirely filled, this metric may help to identify some differences between the modeling approaches even if the flood extent is similarly retrieved.

## 4   Results

Figure 4 illustrates the evaluation results obtained in the case of the Aude 2018 event with the Floodos method. This figure represents both the evaluation against observed flood areas (colored areas) and high water marks (colored points). It shows

an overall good agreement between simulations and observations. Nevertheless, some clusters of large errors are observed locally (zones 2 to 5): these correspond to external sources of errors, which are common to the 3 mapping methods and will be presented in the discussion section (see Sect. 5.2). Zone number 1 corresponds to an area for which the Floodos approach performs significantly better than the two other ones. This case will be discussed in Sect. 5.1.

Dynamic maps enabling a detailed visualisation of the simulation results for all the case studies and mapping methods are 295 provided (Hocini and Payrastre, 2020), see the data availability section. The next sections provide a synthetic analysis of the evaluation results based on the CSI scores computed at the river reach scale, and on the differences between simulated and observed water surface elevations.

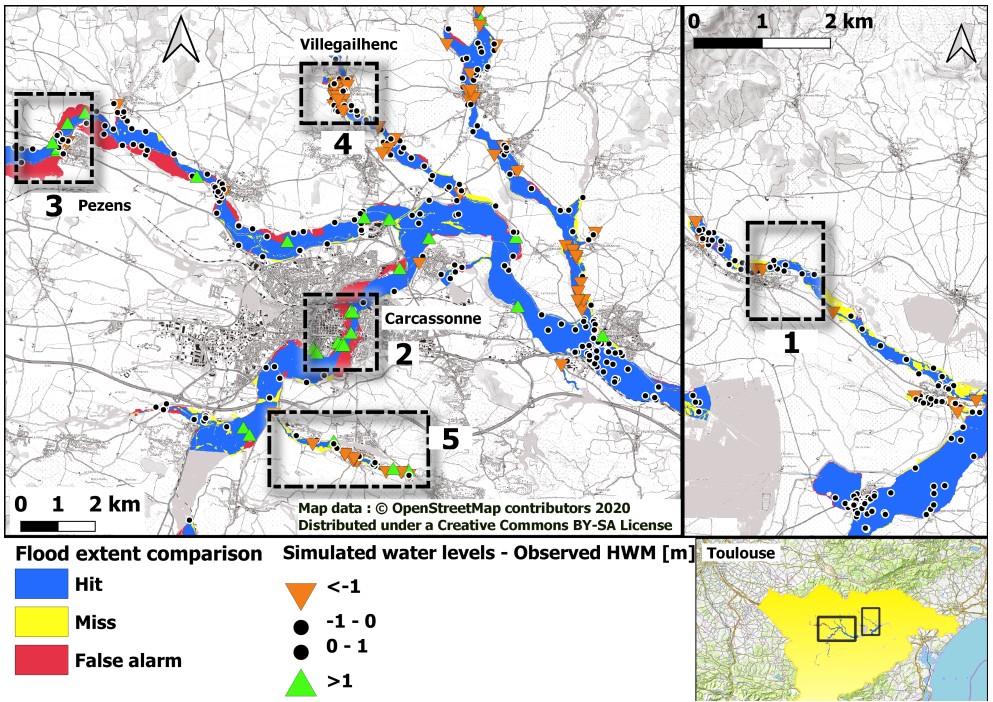

**Figure 4.** Simulated and observed flood areas and water levels, for the Aude 2018 event and the Floodos simulation approach

## 4.1 Simulated flood areas

Figure 5 presents a comparison of the CSI scores obtained for the three mapping methods on the Argens and Aude case studies 300 (observed inundation extent is not available for the Alpes Maritimes case study). This figure shows a clear grading in the ability of the methods to retrieve the extent of the inundated area. Particularly, the HAND/MS method seems to result in significantly lower performance (lower CSIs).

A detailed analysis of Fig. 5.a and c shows that the HAND/MS approach may perform similarly to other approaches in some sections, but that very large errors are observed on specific rivers reaches. These differences may be attributed to the large

level of simplification of the method, particularly: 1 - the fact that riverbed geometry is averaged at the river section scale, 2 - the "boundary" effects between sub-basins which may break the continuity of flow between river sections, particularly at confluences which number is increased here by the level of detail of the river network , and 3 - the absence of representation of backwater effects. However, the discussion section will also illustrate another important cause for these differences (see Sect. 5.1).

Overall similar and satisfactory results are observed for the caRtino 1D and Floodos 2D approaches, with a slight advantage for Floodos, for which the 15% quantiles of CSI values exceed 50 %, and the median CSI values are close to 80%. The largest observed differences between the two methods seem to be concentrated on a limited number of river reaches (Fig. 5.b and d). They are often observed in a context of wide and flat floodplains, sometimes with presence of dikes. In these cases, the complex connection between river bed and floodplain, and the non-uniform flow directions, may limit the validity of 1D approach and
complexify its automatic adaptation in terms of width and orientation of the cross-sections. An example of such a situation is also presented in the discussion section.

Finally, Fig. 5 also shows that the lower CSIs values (below 50%) often occur in the same river sections for the three methods. These low values are mainly related to external error sources, which are not related to the computation method used, but rather to input data (peak discharges, DTM, ..., see Sect. 5.2).

**4.2  Simulated water levels**

The comparison results with high water marks are presented on Fig. 6. This second evaluation includes the Alpes-Maritimes case study. Considering the possible errors on observed HWMs elevation (see section 3.1), simulation errors up to 50 cm may be considered as non significant. However, these error sources are common to the three mapping approaches and should not result in any differences in the results obtained with the three methods.

The results globally confirm the observations made for the inundation extents. Water levels are significantly better simulated with the 2D Floodos model, for which the 70% and 90% error limits shown on the boxplots do not exceed respectively [-0.9 m , +0.7 m] , and [-1.4 m , +1.1 m]. The dispersion of errors is significantly higher with the caRtino 1D method. The HAND/MS method results both in a higher dispersion of errors and a significant negative bias. This may be partly due to the choice of roughness coefficients, since a negative bias is also observed with the two other methods. However, the negative
bias is systematically higher for the HAND/MS method than for the two other ones, suggesting a systematic tendency of the HAND/MS approach to underestimate water levels. An explanation for this phenomenon is presented in the discussion section.

**5  Discussion**

**5.1  Origin of the main differences between the three simulation approaches**

The hierarchy observed is very similar and consistent between the three case studies. It also appears fully consistent with the
level of simplification of the three methods used: logically, the SWE hydraulic methods outperform the HAND/MS approach,

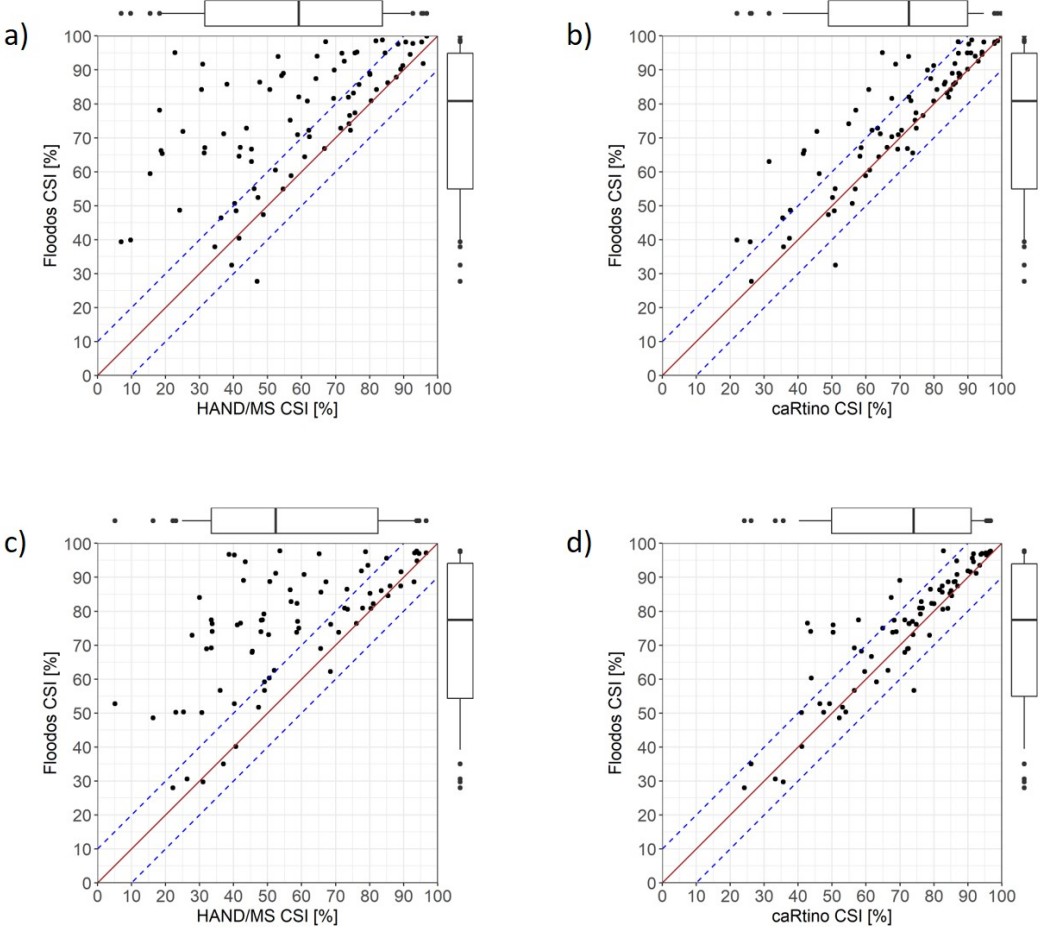

**Figure 5.** Comparison of CSIs computed for the river reaches : a) HAND/MS vs Floodos, Argens 2010 event, b) caRtino 1D vs Floodos, Argens 2010 event, c) HAND/MS vs Floodos, Aude 2018 event, d) caRtino 1D vs Floodos, Aude 2018 event. The boxplots represent respectively the 5% and 95% (whiskers), and the 15% and 85% quantiles (boxes)

and the 2D SWE resolution scheme provides slightly better results than the 1D one, despite the fact that inertial terms are neglected. The results also illustrate that the HAND/MS approach can locally have very similar performance than the two other approaches, but largely fails to retrieve the inundation extent in some cases where the two other approaches perform very well. To a lesser extent, the caRtino 1D approach also shows a significantly lower performance than Floodos for a limited number of reaches (see Fig. 5).

Figure 7 shows an example of a river section for which the three methods lead to significantly different results. This section is located on the Argent-Double river at La Redorte (corresponding to zone 1 on Fig. 4). In this section, the CSI scores are respectively 69% for the Floodos model, 63% for the caRtino 1D model, and 35% for the HANS/MS method. This example illustrates an unexpected limitation of the HAND/MS approach, due to the configuration of the floodplain encountered here:

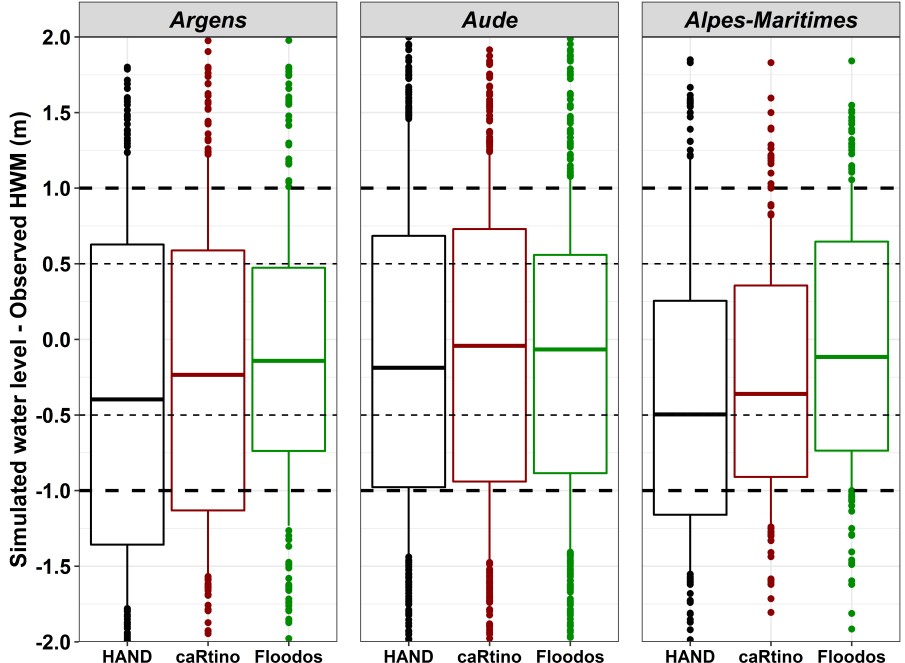

**Figure 6.** Comparison of simulated water levels and observed high water marks (HWM), for the three methods and the three events. The boxplots represent respectively the 5% and 95% (whiskers), and the 15% and 85% quantiles (boxes)

large and flat floodplain on the left hand bank, with a longitudinal slope significantly higher than the transverse slope in the floodplain. In such a situation, a large number of HAND pixels in the floodplain are connected to a drainage point located several hundred meters downstream, with consequently a very large HAND height values. As shown on Fig. 7).d, this results in a large difference between the actual and HAND cross-sectional shapes. The HAND profile shows a sudden increase in HAND elevations which drastically limits the extent of the simulated floodplain (Fig. 7.a). Figure 7.d also shows that on the right bank,

where the transverse slope is largely higher, the shape of the HAND profile is very similar to the actual cross-section. This cross-section retrieval error limits the wetted area and causes an increase and overestimation of the simulated water surface levels in the HAND/MS results for the affected reaches. But this is largely compensated by the under-estimation of water levels in the areas simulated as non-flooded because of the "wall" effect in the HAND profile. The flood extent underestimation effect presented here, due to the unexpected shape of the HAND profile, is frequently observed in the three case studies, which largely

explains the negative bias shown on Fig. 6 and the lower CSIs on Fig. 5 with the HAND/MS method. The two other methods better retrieve the actual flood extent on Fig. 7, but they also show significant differences. Indeed, Fig. 7).b and e show that the 1D approach does not ensure a hydraulic continuity in the floodplain and between the successive river cross-sections. Due to longitudinal variations of these river cross-sections, parts of the floodplain are non-flooded according to the 1D model, because over-bank flow does locally not occur.

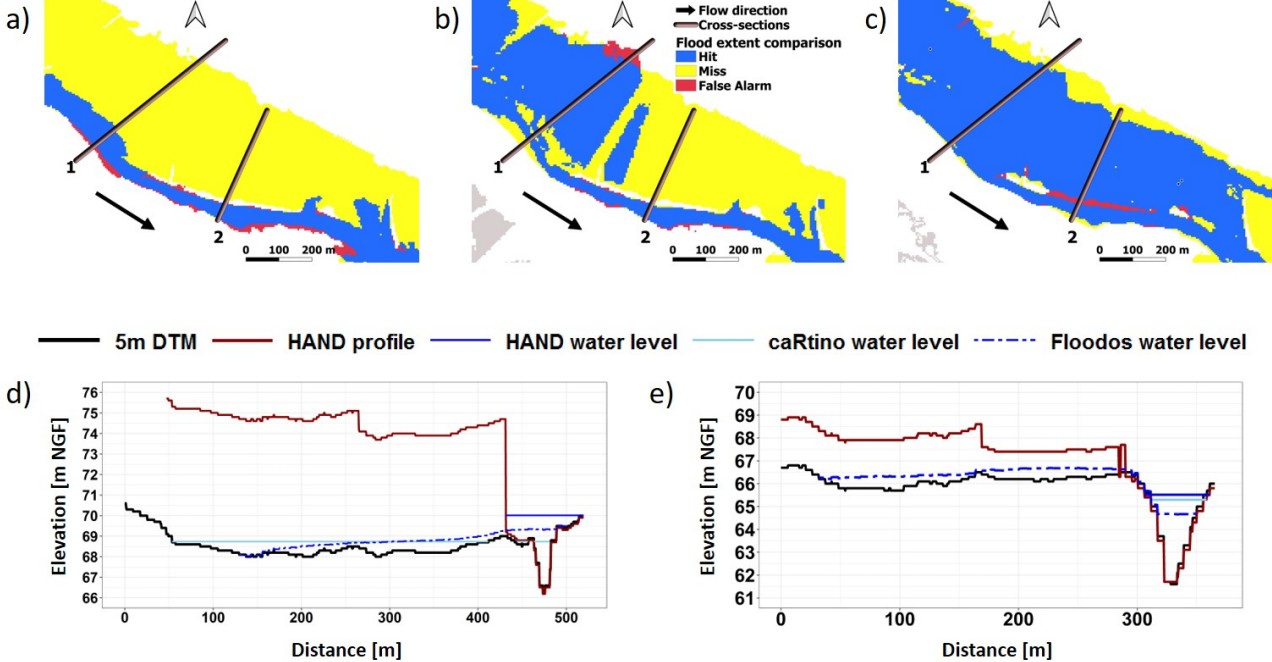

**Figure 7.** Simulation results on the Argent-Double river at Le Redorte, showing large differences between the three tested approaches: a) HAND/MS, b) caRtino 1D, c) Floodos, d) cross-section 1, comparing water surface and terrain elevations, and the HAND profile (addition of the HAND height values and the elevation of the drainage point in the section), e) cross-section 2 (same information as cross-section 1)

## 5.2 Illustration of main error sources affecting all methods

Fig. 4 clearly showed that the larger water level underestimations or over-estimations are spatially clustered. This is observed for the three mapping methods, suggesting that the dominating error sources could be due to input simulation data in these cases (estimated peak discharges or DTM for instance). This section presents four examples of such clusters of errors, corresponding to zones 1 to 4 on Fig. 4. The results presented here were all obtained with the Floodos model.

### 5.2.1 Errors induced by the limitations of the DTM

First, Fig. 8 presents two examples of large water level over-estimations mostly due to imperfections in the terrain input data. The first example (Fig. 8.a and b, zone 2 on Fig. 4) corresponds to the Aude river at Carcassonne. In this section the river bathymetry is significant, and the peak discharge of the flood was close to the limit of the riverbed capacity as no significant inundation was observed. In this case, the absence of bathymetric surveys in the DTM has a significant effect on the retrieved cross-sectional shapes and on the river bed capacity, resulting in a significant increase of the simulated water level, and simulated over-bank flows. The second example (Fig. 8.c and d, zone 3 on Fig. 4) corresponds to the Fresquel river at Pezens. In this section dikes are separating the riverbanks and the floodplain. This is a specificity of the Aude case study, where

numerous flood defense structures have been built, especially along the Fresquel river and in the downstream floodplains of the Aude river. Figure 8.d shows that the relief of the dikes is smoothed out in the 5-m DTM if compared to a higher resolution 1-m DTM. Again in this section, the peak discharge of the flood is close to the capacity of the river bed. The underestimation of the dike crest altitude causes in this case a large overestimation of the flood extent on the right bank of the river.

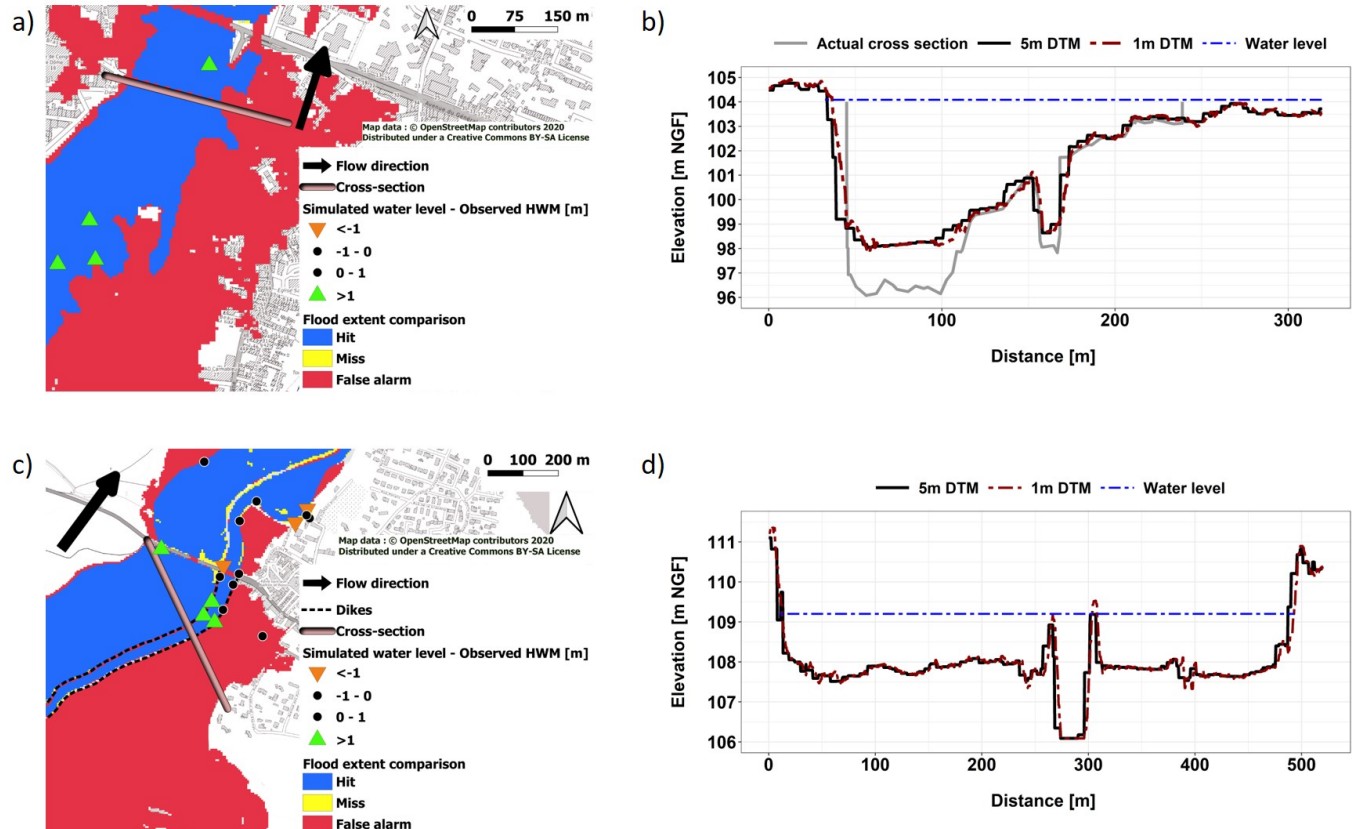

**Figure 8.** Illustration of errors induced by the limitations of the DTM: a) Aude river at Carcassone, simulation results (Floodos model), b) Aude river at Carcassone, cross-sections and simulated water level, c) Fresquel river at Pezens, simulation results (Floodos model), d) Fresquel river at Pezens, cross-sections and simulated water level

### 5.2.2 Local effects of possible bridge blockages or peak discharge errors

On the other hand, large water level underestimations are also observed for some reaches. Two examples are provided on Fig. 9. The first case (Fig. 9.a, zone 4 on Fig. 4), corresponds to the inundation of Villegailhenc village by the Trapel river. In this case, a bridge located in the village has been partly obstructed and submerged during the flood, causing a large backwater effect and very high water levels (>2m) in the vicinity of the bridge. The bridge was finally destroyed by the flood as shown by the picture on Fig. 9.a. Such important backwater effects, often related to bridge blockages, are erratic phenomenons that cannot

be easily forecasted and accounted for in the automatic simulations. They may result in large underestimations of the water levels immediately upstream and downstream the bridges. This situation is encountered at several points in the presented case studies, particularly in sections where the floods were the most intense (estimated return periods often exceeding 100 years). The second example (Fig. 9.c and d, zone 4 on Fig. 4) corresponds to the Fount Guilhen river at Cazilhac. In this case the origin of the underestimation is more difficult to explain. As no clear error appear in the terrain description, the discharge estimations and/or the choice of roughness coefficients may be at the origin of the underestimation of water levels and inundation extent. However, Fig. 9.c shows in this case that a reasonable variation in the peak discharge value (set from 84.2 $m^3.s^{-1}$ to 116.5 $m^3.s^{-1}$ to remain consistent with rainfall observations) is not sufficient to compensate the underestimation effect. Since the selected roughness value (n=0.066) is already relatively high, an underestimation of the locally estimated rainfall intensities is suspected to be at the origin of the errors in this case (Caumont et al., 2020).

## 5.3  Computation times

The computation times required on a single CPU (Intel Core i7-7700 3.60 GHz - 32 Gb RAM) for the three mapping methods are presented in Table 1. They are well correlated with the length of the river network and the number of river reaches, except for the HAND/MS method which is very fast but less predictable. A factor of ten in average is observed between computation times of the HAND/MS and the caRtino 1D models, and of two in average between the caRtino 1D and the Floodos 2D models. As expected, the SWE 2D approach is computationally the most expensive. Butthe computation times remain reasonable for the 5 m resolution used here, and  first parallel computations achieved using a 32 cores and 128 GB RAM cluster suggest that they may still easily be reduced by a factor 4 with the Floodos model. However, the resulting computation times remain large for real time applications, considering the current refreshment frequency of 1h for short-range rainfall nowcasting products.

Another important difference is the relative weight of the pre-treatment phase in the total computation time: pre-treatments are largely preponderant for the HAND/MS and caRtino 1D methods, while only the computation phase is present in the Floodos 2D model. This may be seen as an advantage of HAND/MS and caRtino 1D for real time applications, in which only the computation phase has to be repeated, and hence the required computation times are highly limited for these two methods.

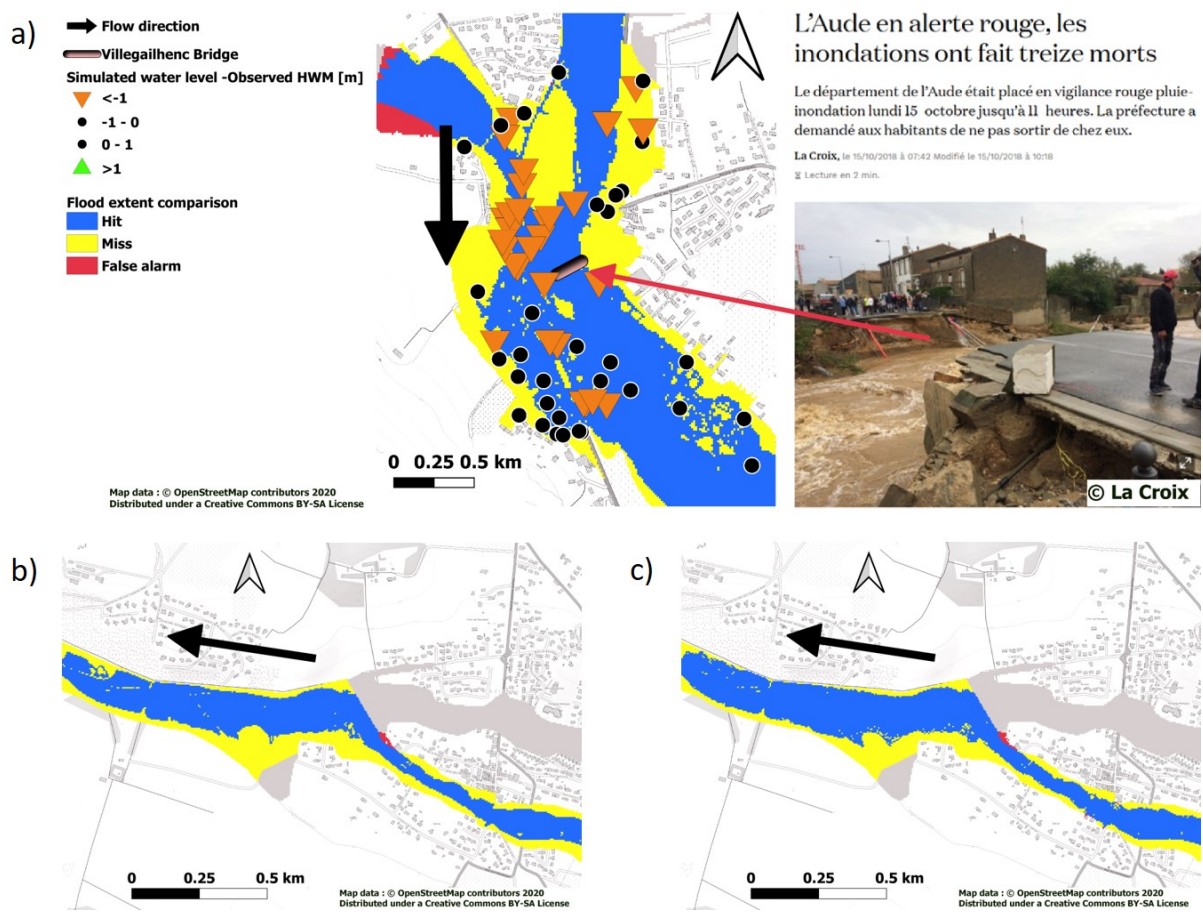

**Figure 9.** Illustration of large water levels underestimations : a) Trapel river at Villegailhenc, simulation results (Floodos model) and picture of the bridge destroyed by the flood, b) Fount Guilhen river at Cazilhac, simulation results (Floodos model) with the initial discharge value (84.2 $m^3.s^{-1}$ corresponding to a CN value of 70 in the Cinecar rainfall-runoff model), c) Fount Guilhen river at Cazilhac, simulation with a modified discharge (116.5 $m^3.s^{-1}$, CN value of 90)

**Table 1.** Comparison of computation times

| | | Nb of river sections | | | Computation times (min) | | | | | |
| | | | | | Total | | | Average per river section | | |
| Case study | Total river length (km) | HAND/MS | caRtino 1D | Floodos 2D | HAND/MS | caRtino 1D | Floodos 2D | HAND/MS | caRtino 1D | Floodos 2D |
|---|---|---|---|---|---|---|---|---|---|---|
| **Argens** | 585 | 531 | 162 | | 13 | 235 | 631 | 0.025 | 1.45 | 3.9 |
| **Aude** | 569 | 446 | 110 | | 22 | 247 | 522 | 0.05 | 2.25 | 4.75 |
| **Alpes Maritimes** | 104 | 130 | 66 | 9 | 19 | 52 | 78 | 0.15 | 0.79 | 1.2 |

# 6 Conclusion and perspectives

The results presented herein illustrate at first the feasability of reasonably accurate flood mapping on small upstream rivers prone to flash flood, based on DTM-based automated approaches. The results presented here are encouraging in terms of quality, with median CSI values close to 80% for the approach based on the Floodos model.

The comparison of the three mapping approaches shows a clear grading of the methods. This result can be explained here by the fact that we do not have only V-shaped valleys with very simple hydraulic features in the presented case studies. The presence of flat floodplains clearly limits the performances of the HAND/MS approach. It induces errors in the retrieval of the river cross-sectional shapes (a "wall" effect), which limit the extent of the simulated inundations. It also limits the performances of the caRtino 1D approach if compared to the 2D Floodos approach, the difference being mainly illustrated by the reconstitution of water levels. A manual adaptation of the width and orientation of cross sections would be necessary here to improve the performance of the 1D approach. Since high progress in computation times has been made with 2D SWE approaches (including Floodos), such approaches now appear compatible with an application at large scales and at high resolutions, while offering significant gain in terms of accuracy.

A detailed sensitivity analysis to the different sources of errors has not been proposed here. However, the largest errors observed seem to be related to external sources (input data) rather than the computation methods. Using an accurate terrain description appears particularly critical. Therefore, a significant increase in quality can still be expected, for instance by using Lidar DTMs at finer resolutions (computations at 1 m resolution would be possible), and also by significant efforts put on appropriate DTM pre-treatments to better represent structures (dikes, buildings, bridges, ..). Inclusion of bathymetric data in the DTMs also appears as an important and challenging issue for the future (Lague and Feldmann, 2020). However, it should be verified that the gains related to input data accuracy are not masked by other sources of uncertainty (Dottori et al., 2013). The sensitivity to roughness values has also to be further investigated for an appropriate representation of uncertainties, and variable roughness values may also be defined depending on land cover (Sampson et al., 2015; Dottori et al., 2016).

Finally, the methods presented here should be of great help to provide realistic inundation scenarios and develop information about possible flash-flood impacts as a support of flash flood risk management policies (Merz et al., 2020; Ritter et al., 2020). However, further work is still needed to integrate these methods into real-time forecasting chains and assess their performance in this context. The errors on discharge forecasts may indeed be dominating the other sources of uncertainties, and the computation times may also be another important limiting factor. Depending on the considered inundation mapping methods, real time computations may be feasible and may improve the representation of flood-wave volumes and flood dynamics at confluences, whereas off-line libraries of inundation scenarios can be generated and sampled in real time (Dottori et al., 2017), which may help representing discharge uncertainties by selecting multiple scenarios (Leedal et al., 2010). The definition of the best real-time computation strategy is even more complex in the case of flash-floods, because of their very fast evolution dynamics. The delay necessary to run and provide forecasts may indeed highly limit the capacity of emergency services to analyse forecasts and adapt their response strategies by reference to inundation scenarios they are prepared for. Finally, an optimal compromise

has probably to be found in the case of flash floods between the accuracy of inundation forecasts and the rapidity of forecast delivery.

*Data availability.* All simulation and evaluation results are integrated in the HyMeX database (http://dx.doi.org/10.6096/mistrals-hymex.1598) and can be downloaded for a detailed visualization (Hocini and Payrastre, 2020).

*Author contributions.* The contributions to this paper are the following: the initial idea was proposed by Nabil Hocini, François Bourgin
and Olivier Payrastre ; the initial version of the paper was written by Olivier Payrastre and Nabil Hocini, with a contribution of Eric Gaume for the analysis of results ; Nabil Hocini achieved the HAND/MS and Floodos simulations, and the common exploitation of all simulation results ; Olivier Payrastre provided the Cinecar rainfall runoff simulations ; François Bourgin contributed to the application of the Cinecar model, and to the HAND/MS and caRtino 1D mapping approaches; Philippe Davy and Dimitri Lague provided help for the application of the Floodos method ; caRtino 1D simulations were prepared by Lea Poinsignon and Frederic Pons.

*Competing interests.* The authors declare they have no competing interests.

*Acknowledgements.* This work is part of the PICS research project (http://pics.ifsttar.fr), funded by the Agence Nationale de la Recherche (agreement no. ANR-17-CE03-0011). Peak discharge data was obtained as part of the HyMeX research program (http://hymex.org), with financial support from the MISTRALS program of the CNRS, and the Ministry of Ecological and Solidarity Transition (DGPR / SCHAPI). Rainfall data was provided by Météo France, DTM data was obtained from IGN.

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
