# Peer review of "Performance of automated methods for flash flood inundation mapping: a comparison of a DTM filling and two hydrodynamic methods"

_Hydrology and Earth System Sciences, 2020_

## Referee Comment (RC1) · Francesco Dottori (Referee) · 22 Dec 2020

The manuscript describes the application of three different modelling approaches to map flash flood hazard in three case studies in South France. The topic is undoubtedly worth of interest, considering the potential for near-real-time applications and the possibility to include flash flood impacts in future applications. The manuscript is well structured and reasonably well written. The authors perform a detailed analysis of the

model results, including the main types of error found, and this gives the reader a comprehensive overview of the strengths and limitations of each method. In my opinion, the paper will be ready for publications after a moderate revision to correct a few issues.

Main points

L203: "The simulations are all run in steady state regime based on estimated flood peak discharges for each river reach. This leads to neglect the possible dynamic effects related to the inundation of floodplains occurring in unsteady flow regimes. This assumption is considered here as reasonable since the width of the floodplains do not exceed several hundred meters, and the volumes of the floodplains remain limited if compared to the volumes of the floods." I think that the limitations due to this modelling choice need be better explained. Based on the results, this seems indeed a reasonable assumption given that there is no general overestimation for the 1D and 2D models. Still, simulating a steady flow regime using peak flow implies an overestimation of total flood volumes, compared to a real flood wave with increasing and receding limbs. As such, this point should be mentioned in the discussion because it might originate errors in case of flood events where flood volumes are small compared to the floodplain extent. Moreover, steady flow simulations have limitations when modelling the interaction of flood waves at confluences. The underlying assumption is that flood peaks are occurring at the same time (a sort of worst-case scenario), while in reality peaks might occur at different times. This point should also be mentioned.

Section 5.3 . The presentation of run times would be even more informative if the authors could make a more quantitative comparison with run-time required to actually set up a real-time flood simulation. Often, reliable weather forecasts of flash flood events are available only few hours in adavance, meaning that a real-time simulation should be available to emergency responders in ,say, 2-3 hours to be effective and helpful. Considering the usual speed-up attainable for 2D hydraulic models (see for instance Neal et al., 2018, https://doi.org/10.1016/j.envsoft.2018.05.011 ) this seems to be feasible objective, provided that the Floodos model can be parallelized in a similar

way. Could you please elaborate a bit on this?

Conclusions: I suggest to elaborate a bit more the discussion on real-time applications, given its importance. In my opinion, real-time applications are meaningful only with the 1D or 2D hydraulic models, which are both able to simulate flood waves in unsteady flow conditions, including the interaction of flood waves with different timings at confluences. On the contrary, if the steady-state approach is deemed appropriate, then an off-line catalogue (similarly to what done by Dottori et al., 2017, https://doi.org/10.5194/nhess-17-1111-2017) would probably be enough. I would be interested in reading the opinion of the authors on this point. In addition, I suggest to mention the possibility of using the described methods to evaluate flash flood impacts (see the recent works by Merz et al., 2020, https://doi.org/10.1029/2020RG000704; and Ritter et al., 2020, https://doi.org/10.1016/j.envint.2019.105375)

Minor issues

The Title is maybe a bit redundant, consider shortening , e.g.: "Performance of automated methods for mapping flash flood hazard: a comparison of hydrodynamic and geomorphologic methods" or something similar

Abstract L13-14: "With these methods, the inundated areas are overall well retrieved..." Here I would suggest replacing the qualitative evaluation with some quantitative metrics, as done for the water levels

L 19 "Flash floods represent a significant part of flood related damages worldwide". Do you have a quantitative assessment of the share of flash flood damages, for instance in France? You might for instance look at the HANZE dataset by Paprotny et al (2018, https://doi.org/10.5194/essd-10-565-2018)

L 37: "For instance, in France it is estimated that a river network of about 100.000 km should be documented for a comprehensive coverage of the small streams". Is there a reference for this statement?

[Figure]

L62-69: This paragraph doesn't read well due to many references and lists of models. Please try to rearrange the information (e.g. I would put first the sentence "All these methods determine a local discharge/height relationship from..." and then "These methods are applied either directly from the DTM for the AutoRoute method...")

L83: "A significant evaluation and validation effort is proposed..." Maybe better rewrite as "A comprehensive evaluation and validation exercise is proposed..."

L88-90 Please replace "first section", "second section" etc with "Section 2", Section 3" etc

Title of Section 2: I'd rather use "description" than "presentation".

L103-104: "A conventionnal Dinf approach is used here instead of the Geonet approach used in GeoFlood." Could you please either specify the difference or provide references for the two approaches? Also, correct the typo (conventional)

Section 2.2: I suggest renaming the approach as CaRtino-1D HECRAS ,given that HECRAS is the actual hydraulic model applied.

L116: please provide a reference for the Mascaret model here (or remove the mention if not relevant for the study)

L125-126: "Its main limits, already identified in previous works, lie in the 1D scheme which may not be adapted in areas with complex hydraulic features". Please name some of these works here.

L 140: "The model has been compared with the widely used 2D LISFLOOD-FP model (Bates et al., 2010), showing equivalent results and faster computation times." Were these tests performed by Davy et al. as well? Please specify also the reduction in computational time as compared with LISFLOOD-FP.

Section 4 L243-248: This paragraph and Figure 4 might be better placed in a separate subsection after subsection 4.1

Figure 4: it is not clear where these two areas are located within the study area, Please add a smaller map of the study area showing the location of the two boxes

Figure 8: Is it simulated water level in panels b-d?

Figures 8 and 9: I assume that you are using Floodos simulations here right? Please specify this in the text and captions

Section 5.2.1: Accounting for protection structures is indeed a major challenge in any large-scale flood risk assessment. Could you tell how much of the study areas is protected by dykes or other defence structures?

L358 typo: feasibility of reasonably accurate

L375 "The sensitivity to roughness values has also to be further investigated for an appropriate representation of uncertainties". Using variable roughness values according to land cover could be an option for future studies. This is actually a standard practice for large-scale flood models (see Sampson et al., 2015, https://doi.org/10.1002/2015WR016954; and Dottori et al., 2016, https://doi.org/10.1016/j.advwatres.2016.05.002)

---

## Referee Comment (RC2) · Paul Bates (Referee) · 6 Jan 2021

This paper tests the ability of three different automatically built flood inundation mapping methods to predict flood extent and high water marks recorded during three extensive flash floods events in French catchments. It differs from much other work in this area that seeks to benchmark inundation modelling methods by virtue of the large spatial scale over which comparisons are undertaken, the large volume of comparison

data (which comprises many thousands of high water marks) and the even treatment of the different methods. Studies in this area can sometimes be undertaken in only localised areas using limited data which cannot discriminate well between competing approaches and papers can also suffer from a kind of 'unconscious bias' towards the researcher's own model. All these pitfalls are avoided in this paper, and the resulting study is therefore a serious one. The research is well executed and mostly very well presented, and I think could be published in HESS with the correction of the following points.

1. I think there should be a bit more discussion of the limitations of using a steady state approximation to model flash floods. I guess this works ok because the automatic model build splits each catchment into small reaches where it is much more plausible to assume steady state conditions, but it would be nice to hear the justification from the authors.

2. Somewhere in the paper there needs to be a discussion about the limitations of using flood extent as a validation metric in narrow valleys and headwater catchments, especially during catastrophic floods events which are very likely to be valley-filling. In these circumstances it may be easy for models to replicate inundation extent and this metric may not be able to effectively discriminate between competing approaches. I have a suspicion that this effect may explain quite a lot about why the performance of the HAND method varies markedly in space.

3. Line 29. Methods are now starting being developed to estimate unknown bathymetry in large catchments which might be worth mentioning here e.g.

Gleason, C. J., & Smith, L. C. (2014). Toward global mapping of river discharge using satellite images and at-many-stations hydraulic geometry. Proceedings of the National Academy of Sciences, 111(13), 4788-4791. https://www.pnas.org/content/pnas/111/13/4788.full.pdf

Grimaldi, S., Li, Y., Walker, J. P., & Pauwels, V. R. N. (2018). Effective Representation of River Geometry in Hydraulic Flood Forecast Models. Water Resources Research, 54(2), 1031-1057. https://agupubs.onlinelibrary.wiley.com/doi/abs/10.1002/2017WR021765

Neal, J. C., Odoni, N. A., Trigg, M. A., Freer, J. E., Garcia-Pintado, J., Mason, D. C., et al. (2015). Efficient incorporation of channel cross-section geometry uncertainty into regional and global scale flood inundation models. Journal of Hydrology, 529, 169-183.

Brêda, J. P. L. F., Paiva, R. C. D., Bravo, J. M., Passaia, O. A., & Moreira, D. M. (2019). Assimilation of Satellite Altimetry Data for Effective River Bathymetry. Water Resources Research, 55(9), 7441-7463. https://agupubs.onlinelibrary.wiley.com/doi/abs/10.1029/2018WR024010

4. Line 51. The key point about the paper by Savage et al quoted here is that they found that below particular grid scales the model precision became spurious 'given other uncertainties'. Might be worth editing to include this idea.

5. Line 2014-5. the sentence starting "A conventional Dinf ..." could do with just a bit more explanation to be understood by a more general audience not familiar with these terms.

6. Line 110. There needs to be a bit more discussion about the limitations of HAND. My understanding of the method is that it assumes that: (i) the water level is uniform over the reach and (ii) that all cells with elevation lower than the water level are inundated even if there is no flowpath connection to the channel. See Figure 1 in this paper https://nhess.copernicus.org/articles/19/2405/2019/nhess-19-2405-2019.pdf. Extended cross section 1D methods can also suffer from the second of these issues. Both assumptions are obviously very different to how floods behave in reality and will explain some of the misprediction with HAND and the 1D model. 2D approaches automatically avoid both issues.

7. Line 117. There are now a few papers on the importance of cross section spacing

in 1D models which you should probably cite here. Would also be worth a sentence discussing how your model build dealt with this issue. See for example: Anuar Md Ali, Giuliano Di Baldassarre & Dimitri P. Solomatine (2015) Testing different cross-section spacing in 1D hydraulic modelling: a case study on Johor River, Malaysia, Hydrological Sciences Journal, 60:2, 351-360, DOI: 10.1080/02626667.2014.889297.

8. Line 141. I think this statement needs a reference.

9. Section 3.1. This section needs to include a more extensive discussion of the uncertainties in the observed data. This then needs to be picked up in the discussion to determine whether the models can match the observed data to within error or not. You already discuss the terrain data error in the paper, but don't really say much about errors in the observed discharge other than the rainfall-runoff model generally matched the observed discharge to within 10%. However, discharge gauging during extreme floods is fraught with difficulty and you need to consider the likely error in this, even if this can only be a best estimate made with reference to other studies. The error in the rainfall-runoff model is somewhat misleading, as most such models usually have enough degrees of freedom to be able to match 'observed' data adequately, even if it has error and is disinformative. I would expect discharge gauging during flash floods to be have errors of at least +-20%. Similarly, you need to say a lot more about how the high water marks were collected, what their likely error are and what QA/QC procedures you undertook to clean up these data. I think for each catchment you should include a plot of HWMs versus thalweg distance and also plot on this the overall valley slope derived from LiDAR data. This will show if there are obvious outliers as we would expect flood water surface profiles to decrease slowly and monotonically in a downstream direction. Regions of supercritical flow may be an exception to this rule of thumb, but, in general, this is the pattern we might expect. Plotting the model water surface slopes on these graphs would also be very informative. A further quality check is to plot the difference in elevation between pairs of high water marks and compare this to the valley slope. Figure 3 in this paper might be a useful template:

Fewtrell, T.J., Neal, J.C., Bates, P.D. and Harrison, P.J. (2011). Geometric and structural river channel complexity and the prediction of urban inundation. Hydrological Processes, 25, 3173-3186. (10.1002/hyp.8035).

Lastly, you need to say a lot more about how the inundation extent was mapped and what were the likely errors in this.

10. Section 3.2. It would be good to include a plot of the hydrographs so readers can better appreciate the event dynamics. Did the events lead to any regions of supercritical flow and, if so, how well do you expect the models to perform at these locations?

11. Line 238. This threshold of 65% is arbitrary. I don't disagree with it, but I think you need to better justify its choice.

12. Line 241: What did you do about calculating the error at high wate marks where the model did not predict any inundation?

13. Line 250. I tried to download the data but the site did not allow me to do this. Please can you fix this.

14. Section 4: Through this section and the discussion you need to take into account the impact of the observed data errors on your ability to make inferences and discriminate between the three models. Where do the models match the data to within error and where do they not?

15. Line 334. Was there any supercritical flow at the bridge blockages?

16. Line 373. Here you might want to return to the point about the push to finer resolution becoming spurious at particular scales given other uncertainties. See the excellent paper by my co-reviewer Francesco Dottori that first identified this issue:

Dottori, F., Di Baldassarre, G., and Todini, E. (2013), Detailed data is welcome, but with a pinch of salt: Accuracy, precision, and uncertainty in flood inundation modeling, Water Resour. Res., 49, 6079– 6085, doi:10.1002/wrcr.20406.

17. Line 378. One solution to the Near Real Time prediction problem with 2D models is to pre-compute a library of inundation simulations at different flow rates which can then be sampled extremely quickly based on the observed or forecast discharge. The advantage of this approach is that one can sample a range of flows of weight them according to their likelihood, thereby also accounting either for the uncertainty in the observed data or in the forecast. See the following paper by David Leedal for some ideas on how to do this:

Leedal, D., Neal, J., Beven, K., Younger, P. and Bates, P. (2010). Visualisation approaches for communicating real-time flood forecasting level and inundation information. Journal of Flood Risk Management, 3 (2), 140-150. (10.1111/j.1753-318X.2010.01063.x).

I hope these comments are useful and I very much look forward to seeing the paper in print.

Paul Bates

University of Bristol

---

## Author Comment (AC1) · 23 Feb 2021

At first we would like to warmly thank both referees for their careful reading of the manuscript, their positive appraisal and their very useful comments and suggestions to improve the overall quality of the manuscript. We provide hereafter a detailed answer explaining how we managed to address each of the raised issues.

**Francesco Dottori (Referee)**
francesco.dottori@ec.europa.eu

The manuscript describes the application of three different modelling approaches to map flash flood hazard in three case studies in South France. The topic is undoubtedly worth of interest, considering the potential for near-real-time applications and the possibility to include flash flood impacts in future applications. The manuscript is well structured and reasonably well written. The authors perform a detailed analysis of the model results, including the main types of error found, and this gives the reader a comprehensive overview of the strengths and limitations of each method. In my opinion, the paper will be ready for publications after a moderate revision to correct a few issues.

Main points
L203: "The simulations are all run in steady state regime based on estimated flood peak discharges for each river reach. This leads to neglect the possible dynamic effects related to the inundation of floodplains occurring in unsteady flow regimes. This assumption is considered here as reasonable since the width of the floodplains do not exceed several hundred meters, and the volumes of the floodplains remain limited if compared to the volumes of the floods." I think that the limitations due to this modelling choice need be better explained. Based on the results, this seems indeed a reasonable assumption given that there is no general overestimation for the 1D and 2D models. Still, simulating a steady flow regime using peak flow implies an overestimation of total flood volumes, compared to a real flood wave with increasing and receding limbs. As such, this point should be mentioned in the discussion because it might originate errors in case of flood events where flood volumes are small compared to the floodplain extent. Moreover, steady flow simulations have limitations when modelling the interaction of flood waves at confluences. The underlying assumption is that flood peaks are occurring at the same time (a sort of worst-case scenario), while in reality peaks might occur at different times. This point should also be mentioned.

This initial formulation indeed does not provide much details on the limitations related to the steady state assumption. We propose to replace this with a more developed description: "The simulations are all run in steady state regime based on estimated flood peak discharges for each river reach. The steady state assumption may lead to an overestimation of the inundation extent and depths if the volume of the flood wave is limited in comparison with the storage capacity of the floodplain. This assumption is considered here as reasonable since the widths of the floodplains do not exceed several hundred meters, and therefore the corresponding floodplain storage capacities should remain limited. The computation based on flood peak discharges may also lead to an overestimation of backwater effects at confluences, because of the underlying assumption that maximum peak discharges occur simultaneously for all river branches at a confluence. Lastly, the variations of peak discharges along each river reach are not represented, but these variations are limited since the delineated river reaches have a limited length"

Section 5.3. The presentation of run times would be even more informative if the authors could make a more quantitative comparison with run-time required to actually set up a real-time flood simulation. Often, reliable weather forecasts of flash flood events are available only few hours in adavance, meaning that a real-time simulation should be available to emergency responders in ,say, 2-3 hours to be effective and helpful. Considering the usual speed-up attainable for 2D hydraulic models (see for instance Neal et al., 2018, https://doi.org/10.1016/j.envsoft.2018.05.011 ) this seems to be feasible objective, provided that the Floodos model can be parallelized in a similar way. Could you please elaborate a bit on this?

We propose here to add a sentence to illustrate to which extent the computation times may be reduced by using parallel computing, and provide a comparison with the current update frequency of short range rainfall nowcasts : "As expected, the SWE 2D approach is computationally the most expensive. But the computation times remain reasonable for the 5 m resolution used here, and first parallel computations achieved using a 32 cores and 128 GB RAM cluster suggest that they may still easily be reduced by a factor 4 with the Floodos model. However, the resulting computation times remain large for real time applications, considering the current refreshment frequency of 1h for short-range rainfall nowcasting products. "

Conclusions: I suggest to elaborate a bit more the discussion on real-time applications, given its importance. In my opinion, real-time applications are meaningful only with the 1D or 2D hydraulic models, which are both able to simulate flood waves in unsteady flow conditions, including the interaction of flood waves with different timings at confluences. On the contrary, if the steady-state approach is deemed appropriate, then an off-line catalogue (similarly to what done

by Dottori et al., 2017, https://doi.org/10.5194/nhess-17-1111-2017) would probably be enough. I would be interested in reading the opinion of the authors on this point. In addition, I suggest to mention the possibility of using the described methods to evaluate flash flood impacts (see the recent works by Merz et al., 2020, https://doi.org/10.1029/2020RG000704; and Ritter et al., 2020, https://doi.org/10.1016/j.envint.2019.105375)

We agree that questions related to real time applications and representation of FF impacts are of great importance. We therefore developed the last paragraph of the conclusion in the following way: " Finally, the methods presented here should be of great help to provide realistic inundation scenarios and develop information about possible flash-flood impacts as a support of flood risk management policies (Merz et al., 2020; Ritter et al., 2020). However, further work is still needed to integrate these methods into real-time forecasting chains and assess their performance in this context. The errors on discharge forecasts may indeed be dominating the other sources of uncertainties, and the computation times may also be another important limiting factor. Depending on the considered inundation mapping methods, real time computations may be feasible and may improve the representation of flood-wave volumes and flood dynamics at confluences, whereas off-line libraries of inundation scenarios can be generated and sampled in real time (Dottori et al., 2017), which may help representing discharge uncertainties by selecting multiple scenarios (Leedal et al., 2010). The definition of the best real-time computation strategy is even more complex in the case of flash-floods, because of their very fast evolution dynamics. The delay necessary to run and provide forecasts may indeed highly limit the capacity of emergency services to analyse forecasts and adapt their response strategies by reference to inundation scenarios they are prepared for. Finally, an optimal compromise has probably to be found in the case of flash floods between the accuracy of inundation forecasts and the rapidity of forecast delivery."

Minor issues
The Title is maybe a bit redundant, consider shortening , e.g.: "Performance of automated methods for mapping flash flood hazard: a comparison of hydrodynamic and geomorphologic methods" or something similar

The title has been modified as follows: "Performance of automated methods for flash flood inundation mapping: a comparison of a DTM filling and two hydrodynamic methods"

Abstract L13-14: "With these methods, the inundated areas are overall well retrieved..." Here I would suggest replacing the qualitative evaluation with some quantitative metrics, as done for the water levels
We replaced the sentence with "With these methods, a good retrieval of the inundated areas is illustrated by Critical Success Index median values close to 80%, and …"

L 19 "Flash floods represent a significant part of flood related damages worldwide". Do you have a quantitative assessment of the share of flash flood damages, for instance in France? You might for instance look at the HANZE dataset by Paprotny et al (2018, https://doi.org/10.5194/essd-10-565-2018)
A reference to a report edited by CCR (Caisse Centrale de Réassurance) on natural disasters in France since 1982 has been added: "For instance, in France eight floods caused insurance losses exceeding 500 million euros over the period 1989-2018, among which 4 were flash floods (CCR, 2019)".

L 37: "For instance, in France it is estimated that a river network of about 100.000 km should be documented for a comprehensive coverage of the small streams". Is there a reference for this statement?
We added an explanation for this statement: "For instance, in France the entire stream network includes 120.000 km of rivers of more than 1m width, whereas flood hazard information is concentrated on the 23.000 km of main rivers, corresponding to the network covered by the Vigicrues national flood forecasting service. It can thus be estimated that about 100.000 km of small rivers should be documented with hazard information to ensure a comprehensive coverage."

L62-69: This paragraph doesn't read well due to many references and lists of models. Please try to rearrange the information (e.g. I would put first the sentence "All these methods determine a local discharge/height relationship from..." and then "These methods are applied either directly from the DTM for the AutoRoute method...")
We reformulated this paragraph in the suggested manner: "Direct DTM filling approaches have been developed more recently. All these methods are based on a local discharge/water height relationship determined from i) the cross-section and longitudinal profile geometries, and ii) a local hydraulic formula: Manning-Strickler (ZhengXing et al., 2018; Zheng at al., 2018; Johnson et al. 2019, GarousiNejad et al., 2019} or Debord (Rebolho et al., 2018). The cross-section geometry is either extracted locally from the DTM for the AutoRoute method (Follum et al., 2017, 2020), or averaged at the river reach scale based on a Height Above Nearest Drainage raster (Nobre et al., 2011) for the following methods : f2HAND (Speckhann et al., 2017); Geoflood (Zheng et al., 2018); MHYST (Rebolho et al., 2018); Hydrogeomorphic FHM (TavaresdaCosta et al., 2019). "

L83: "A significant evaluation and validation effort is proposed..." Maybe better rewrite as "A comprehensive evaluation and validation exercise is proposed..."
The sentence has been reformulated as suggested.

L88-90 Please replace "first section", "second section" etc with "Section 2", Section 3"etc
The replacements have been made.

Title of Section 2: I'd rather use "description" than "presentation".
The replacement has been made.

L103-104: "A conventionnal Dinf approach is used here instead of the Geonet approach used in GeoFlood." Could you please either specify the difference or provide references for the two approaches? Also, correct the typo (conventional)
A reference has been added for the Dinf approach: Tarboton, D. G., (1997), A New Method for the Determination of Flow Directions and Contributing Areas in Grid Digital Elevation Models, Water Resources Research, 33(2): 309-319. The Geotnet approach is described in Zheng et al (2018).

Section 2.2: I suggest renaming the approach as CaRtino-1D HECRAS ,given that HECRAS is the actual hydraulic model applied.
The approach has been renamed  as suggested

L116: please provide a reference for the Mascaret model here (or remove the mention if not relevant for the study)
A reference has been added: MASCARET : a 1-D Open-Source Software for Flow Hydrodynamic and Water Quality in Open Channel Networks, N. Goutal, J.-M. Lacombe, F. Zaoui and K. El-Kadi-Abderrezzak, River Flow 2012 – Murillo (Ed.), pp. 1169-1174, doi:10.1201/b13250

L125-126: "Its main limits, already identified in previous works, lie in the 1D scheme which may not be adapted in areas with complex hydraulic features". Please name some of these works here.
A reference to Le Bihan et al. (2017) has been added here: https://doi.org/10.5194/hess-21-5911-2017

L 140: "The model has been compared with the widely used 2D LISFLOOD-FP model (Bates et al., 2010), showing equivalent results and faster computation times." Were these tests performed by Davy et al. as well? Please specify also the reduction in computational time as compared with LISFLOOD-FP.
Yes the comparison is included in Davy et al. (2017), but according to the authors this comparison should not be considered as a benchmark since their use of the LISFLOOD-FP may still be optimized. Thus it is thus difficult to conclude on the reduction level of computation times. This has been more explicitly mentioned: "Davy et al. (2017) indicate the CPU time changes approximately linearly with the number of pixels of the computation domain. They compared Floodos with the widely used 2D LISFLOOD-FP model (Bates et al., 2010). They obtained similar results and faster computation times with Floodos, although they mention this comparison should not be considered as a benchmark."

Section 4 L243-248: This paragraph and Figure 4 might be better placed in a separate subsection after subsection 4.1
Figure 4 provides a generic illustration of both evaluations of simulated flood areas (subsection 4.1) and simulated water levels (subsection 4.2). Therefore we prefer to present it at the beginning of section 4.

Figure 4: it is not clear where these two areas are located within the study area, Please add a smaller map of the study area showing the location of the two boxes
We updated the figure to explicitly indicate the location of the two presented areas.

Figure 8: Is it simulated water level in panels b-d?
Yes indeed, this has been mentioned in the figure captions.

Figures 8 and 9: I assume that you are using Floodos simulations here right? Please specify this in the text and captions
Yes indeed, this has been specified in the captions and the text.

Section 5.2.1: Accounting for protection structures is indeed a major challenge in any large-scale flood risk assessment. Could you tell how much of the study areas is protected by dykes or other defence structures?
A sentence has been added to provide this information: "This is a specificity of the Aude case study, where numerous flood defense structures have been built, especially along the Frequel river and in the downstream floodplains of the Aude river".

L358 typo: feasibility of reasonably accurate

This has been corrected.

L375 "The sensitivity to roughness values has also to be further investigated for an appropriate representation of uncertainties". Using variable roughness values according to land cover could be an option for future studies. This is actually a standard practice for large-scale flood models (see Sampson et al., 2015, https://doi.org/10.1002/2015WR016954; and Dottori et al., 2016, https://doi.org/10.1016/j.advwatres.2016.05.002) This possibility has been explicitly mentioned "The sensitivity to roughness values has also to be further investigated for an appropriate representation of uncertainties, and variable roughness values may also be defined depending on land cover (Sampson et al., 2015 ; Dottori et al., 2016)."

---

## Author Comment (AC2) · 23 Feb 2021

At first we would like to warmly thank both referees for their careful reading of the manuscript, their positive appraisal and their very useful comments and suggestions to improve the overall quality of the manuscript. We provide hereafter a detailed answer explaining how we managed to address each of the raised issues.

**Paul Bates (Referee)**
paul.bates@bristol.ac.uk

This paper tests the ability of three different automatically built flood inundation mapping methods to predict flood extent and high water marks recorded during three extensive flash floods events in French catchments. It differs from much other work in this area that seeks to benchmark inundation modelling methods by virtue of the large spatial scale over which comparisons are undertaken, the large volume of comparison data (which comprises many thousands of high water marks) and the even treatment of the different methods. Studies in this area can sometimes be undertaken in only localised areas using limited data which cannot discriminate well between competing approaches and papers can also suffer from a kind of 'unconscious bias' towards the researcher's own model. All these pitfalls are avoided in this paper, and the resulting study is therefore a serious one. The research is well executed and mostly very well presented, and I think could be published in HESS with the correction of the following points.

1. I think there should be a bit more discussion of the limitations of using a steady state approximation to model flash floods. I guess this works ok because the automatic model build splits each catchment into small reaches where it is much more plausible to assume steady state conditions, but it would be nice to hear the justification from the authors. Indeed the limitations of using a steady-state assumption were mentioned but insufficiently developed in the initial version of the manuscript. We therefore added the following development in section 3.3: "The simulations are all run in steady state regime based on estimated flood peak discharges for each river reach. The steady state assumption may lead to an overestimation of the inundation extent and depths if the volume of the flood wave is limited in comparison with the storage capacity of the floodplain. This assumption is considered here as reasonable since the widths of the floodplains do not exceed several hundred meters, and therefore the corresponding floodplain storage capacities should remain limited. The computation based on flood peak discharges may also lead to an overestimation of backwater effects at confluences, because of the underlying assumption that maximum peak discharges occur simultaneously for all river branches at a confluence. Lastly, the variations of peak discharges along each river reach are not represented, but these variations are limited since the delineated river reaches have a limited length"

2. Somewhere in the paper there needs to be a discussion about the limitations of using flood extent as a validation metric in narrow valleys and headwater catchments, especially during catastrophic floods events which are very likely to be valley-filling. In these circumstances it may be easy for models to replicate inundation extent and this metric may not be able to effectively discriminate between competing approaches. I have a suspicion that this effect may explain quite a lot about why the performance of the HAND method varies markedly in space.
We fully agree with this. A sentence has been added in section 3.4.1 to remind this important point: "A possible drawback of this metric is that observations of actual flood extents are generally gathered for major floods events, with the objective to establish historical references as support of flood risk management policies. These flood events are likely to be valley-filling, which is clearly the case for the three events considered here. This makes the retrieval of the flood extent much easier to achieve with modeling tools, and may mask the differences of performance between the different competing approaches".

A sentence has also been added in section 3.4.2 to mention that the metric based on water levels helps to compensate this limitation of the Critical Success Index: "In situations where the geomorphologic floodplain is entirely filled, this metric may help to identify some differences between the modeling approaches even if the flood extent is similarly retrieved."

3. Line 29. Methods are now starting being developed to estimate unknown bathymetry in large catchments which might be worth mentioning here e.g. Gleason, C. J., & Smith, L. C. (2014). Toward global mapping of river discharge using satellite images and at-many-stations hydraulic geometry. Proceedings of the National Academy of Sciences, 111(13), 4788-4791. https://www.pnas.org/content/pnas/111/13/4788.full.pdf Grimaldi, S., Li, Y., Walker, J. P., & Pauwels, V. R. N. (2018). Effective Representation of River Geometry in Hydraulic Flood Forecast Models. Water Resources Research, 54(2), 1031-1057. https://agupubs.onlinelibrary.wiley.com/doi/abs/10.1002/2017WR021765. Neal, J. C., Odoni, N. A., Trigg, M. A., Freer, J. E., Garcia-Pintado, J., Mason, D. C., et al. (2015). Efficient incorporation of channel cross-section geometry uncertainty into regional and global scale flood inundation models. Journal of Hydrology, 529, 169-183. Brêda, J. P. L. F., Paiva, R. C. D., Bravo, J. M., Passaia, O. A., & Moreira, D. M. (2019). Assimilation of Satellite Altimetry Data for Effective River Bathymetry. Water Resources Research, 55(9), 7441-7463. https://agupubs.onlinelibrary.wiley.com/doi/abs/10.1029/2018WR024010

We added a sentence in this section to mention these recent advances in unknown river bathymetry estimation: "Even if information on bathymetry is still rarely available, recent advances have been achieved in estimating unknown bathymetry or river channel geometry based on remote sensing or local at-site surveyed data (Gleason et al., 2014; Neal et al., 2015; Grimaldi et al., 2018; Brêda et al, 2019)"

4. Line 51. The key point about the paper by Savage et al quoted here is that they found that below particular grid scales the model precision became spurious 'given other uncertainties'. Might be worth editing to include this idea.

The sentence has been reformulated to better reflect this idea: "For instance, Savage et al. (2016) consider that resolutions finer that 50 m offer a limited gain due to other sources of uncertainties, while resulting in a large increase of computational expense;.."

5. Line 2014-5. the sentence starting "A conventional Dinf . . ." could do with just a bit more explanation to be understood by a more general audience not familiar with these terms.

We reformulated and added a reference for detailed explanations on the Dinf approach: "A conventional approach based on Dinf flow directions (Tarbotton, 1997) is used here instead of .."

Tarboton, D. G., (1997), A New Method for the Determination of Flow Directions and Contributing Areas in Grid Digital Elevation Models, Water Resources Research, 33(2): 309-319.

6. Line 110. There needs to be a bit more discussion about the limitations of HAND. My understanding of the method is that it assumes that: (i) the water level is uniform over the reach and (ii) that all cells with elevation lower than the water level are inundated even if there is no flowpath connection to the channel. See Figure 1 in this paper https://nhess.copernicus.org/articles/19/2405/2019/nhess-19-2405-2019.pdf. Extended cross section 1D methods can also suffer from the second of these issues. Both assumptions are obviously very different to how floods behave in reality and will explain some of the misprediction with HAND and the 1D model. 2D approaches automatically avoid both issues.

The sentence describing the main limitations of the HAND approach has been developed in the following way: "However it is based on several important assumptions. First, the cross-sectional geometry and water level are averaged and supposed to be uniform for each river reach. Therefore, backwater effects due to longitudinal slope and cross section shape variations along one river reach, and/or due to downstream limit conditions, are not represented. Second, longitudinal discharge variations along each river reach cannot be accounted for. Third, the inundation depth at each point of the floodplain depends only on its relative elevation above its nearest downstream drainage point (i.e. the HAND raster value), independently of the real hydraulic connections. This may result in discontinuities : neighbour pixels having similar elevations may be related to different drainage points and hence be attributed different hand values. This is particularly true in the case of flat and wide floodplains and at confluences where neighbour pixels may be connected to different river reaches. In this latter case, the water levels considered for the inundation mapping will also be different for the two neighbour points. "

7. Line 117. There are now a few papers on the importance of cross section spacing in 1D models which you should probably cite here. Would also be worth a sentence discussing how your model build dealt with this issue. See for example: Anuar Md Ali, Giuliano Di Baldassarre & Dimitri P. Solomatine (2015) Testing different cross-section spacing in 1D hydraulic modelling: a case study on Johor River, Malaysia, Hydrological Sciences Journal, 60:2, 351-360, DOI: 10.1080/02626667.2014.889297.

The spacing between cross sections is optimized in caRtino to limit as far as possible the distance between cross sections and avoid in the same time cross section overlapping. The following sentences have been added to explain how it is achieved: "Since the distances between cross-sections may have a significant impact on 1D hydraulic simulation results (Ali et al., 2014), the cross-sections are positioned with the double objective to limit their spacing and avoid overlapping. This is achieved in the following way: i) a constant distance between cross-sections is first used (50 meters in this application) ; ii) a first hydraulic run is conducted to estimate the width of the floodplain ; iii) the distance between cross-sections is then set to a proportion of the floodplain width (here 30%), and the cross-sections are reoriented if crossing each other."

8. Line 141. I think this statement needs a reference.

The comparison with LISFLOOD is presented in Davy et al. (2017), but according to the authors this comparison should not be considered as a benchmark. This has been more explicitly mentioned: "Davy et al. (2017) indicate the CPU time changes approximately linearly with the number of pixels of the computation domain. They compared Floodos with the widely used 2D LISFLOOD-FP model (Bates et al., 2010). They obtained similar results and faster computation times with Floodos, although they mention this comparison should not be considered as a benchmark."

9. Section 3.1. This section needs to include a more extensive discussion of the uncertainties in the observed data. This then needs to be picked up in the discussion to determine whether the models can match the observed data to within error or not. You already discuss the terrain data error in the paper, but don't really say much about errors in the observed discharge other than the rainfall-runoff model generally matched the observed discharge to within 10%. However, discharge gauging during extreme floods is fraught with difficulty and you need to consider the likely error in this, even if this can only be a best estimate made with reference to other studies. The error in the rainfall-runoff model is somewhat misleading, as most such models usually have enough degrees of freedom to be able to match 'observed' data adequately, even if it has error and is disinformative. I would expect discharge gauging during flash floods to be have errors of at least +-20%. Similarly, you need to say a lot more about how the high water marks were collected, what their likely error are and what QA/QC procedures you undertook to clean up these data. I think for each catchment you should include a plot of HWMs versus thalweg distance and also plot on this the overall valley slope derived from LiDAR data. This will show if there are obvious outliers as we would expect flood water surface profiles to decrease slowly and monotonically in a downstream direction. Regions of supercritical flow may be an exception to this rule of thumb, but, in general, this is the pattern we might expect. Plotting the model water surface slopes on these graphs would also be very informative. A further quality check is to plot the difference in elevation between pairs of high water marks and compare this to the valley slope. Figure 3 in this paper might be a useful template: Fewtrell, T.J., Neal, J.C., Bates, P.D. and Harrison, P.J. (2011). Geometric and structural river channel complexity and the prediction of urban inundation. Hydrological Processes, 25, 3173-3186. (10.1002/hyp.8035). Lastly, you need to say a lot more about how the inundation extent was mapped and what were the likely errors in this.

The errors on post flood discharge data have been estimated and are plotted on figure 3, they are indeed mostly close to a +-20% range, but sometimes higher. The caption of figure 3 has been updated to indicate that error bars are plotted. The sentence on rainfall runoff simulation results has also been updated to mention more explicitly the possible remaining errors on simulation results: "Overall, the differences between simulated and observed peak discharges do not exceed +-20\% (see Fig.3). However, observations are mainly based on post-flood surveys and may have large uncertainties, as indicated by error bars on Fig.3. Moreover, observations are not available at each branch of the considered river networks. Therefore, the simulated peak discharges obtained from the rainfall-runoff model may locally differ significantly from the actual ones.".

The HWM data was extracted from the french national HWM database (https://www.reperesdecrues.developpement-durable.gouv.fr). This data is systematically checked before incorporation in the database. Presenting longitudinal profiles for our case studies is difficult because of the length and large ramifications of the considered river networks. Additionally, significant contrasts are locally observed between the HWM reported in the floodplain and the HWM in the main stream at a given location (presence of weirs, waterfalls, ..). The association of HWM to a talweg longitudinal distance may therefore raise some interpretation difficulties. However the graphs proposed by Fewtrell et al. (2011) are easier to plot and are presented below (figure 1) for the 3 case studies. The averaged slopes of the river reaches where HWM are available are also plotted (maximum slope value, and 90% and 50% quantiles for each case study). According to these figures, some HWM elevation differences may indeed appear as outliers since exceeding the maximum slope of river reaches. But this is observed only for a limited number of HMW couples (3% for the Aude , 2% for the Argens, and less than 1% for the Alpes Maritimes case study). And these large elevation differences cannot be systematically attributed to errors in the data, but also to local phenomena such as bridge blockages, presence of weirs or waterfalls. Considering the limited number of HWMs concerned we think that the possible presence of errors does not significantly affect our comparison results.

[Figure]

Figure 1. Comparison of high water marks elevation differences and river reaches averaged slopes (maximum value, 90% and 50 % quantiles): a) Aude case study, b) Argens case study, c) Alpes Maritimes case study.

We therefore chose to mention clearly the origin of the data in section 3.1 and the possible presence of remaining errors in the HWM data: "The HWM data was extracted from the french national HWM database (https://www.reperesdecrues.developpement-durable.gouv.fr). This data is systematically checked before incorporation in the database and therefore should not include large errors. However, errors up to 50 cm should be considered as common considering the accuracy of topographic surveys (HMW location and elevation), and/or possible inappropriate choice of HWMs locations (increase of water surface elevation in front of obstacles, capillary rise of moisture in walls, ..). Some larger errors may also remain for a very limited number of HWMs, and may result locally in large estimated simulation errors. But all these error sources are common to the 3 methods and should not affect the comparison results."

We also added a sentence in section 3.1 to mention the origin of inundation extent observed data: "The detailed mapping of inundation extents, available for the Argens 2010 and Aude 2018 events, was achieved by local authorities based on field surveys in the weeks following the floods. This data should have a good accuracy even if it may have been locally interpolated between field observation points."

10. Section 3.2. It would be good to include a plot of the hydrographs so readers can better appreciate the event dynamics. Did the events lead to any regions of supercritical flow and, if so, how well do you expect the models to perform at these locations?
Observed hydrographs are only seldom available for the considered events (see red crosses on fig.3) since a large part of the gauging stations were destroyed during the floods, particularly for the Argens 2010 and Alpes Maritimes 2015 events. It would be possible to add only one figure to illustrate the (very fast) dynamics of the Aude 2018 flood.
Since the river bed slopes remain limited in a large part the considered river networks, supercritical flows should be observed only very locally.

11. Line 238. This threshold of 65% is arbitrary. I don't disagree with it, but I think you need to better justify its choice.
The threshold of 65% was proposed by Fleishmann et al. (2019). We modified the formulation in the following way to better justify this choice: "Since this metric cumulates overestimation (b) and underestimation (c), it may decrease significantly even for simulation results which appear visually to fit well the observations. Fleishmann et al. (2019) consider that hydrodynamic models with CSI scores greater than 65% at reach scale show satisfactory results."

12. Line 241: What did you do about calculating the error at high water marks where the model did not predict any inundation?
An explanation has been added on this point: "If the model does not predict any inundation at the position of the high water mark, it is considered that the predicted water height is 0 m, and thus the computed error corresponds to the elevation of the high water mark above ground."

13. Line 250. I tried to download the data but the site did not allow me to do this. Please can you fix this.
We checked this point and the downloading seems to be working properly (the "Direct access to public data" button must be used to download the data). We suspect that the problem encountered may depend on the navigator used.

14. Section 4: Through this section and the discussion you need to take into account the impact of the observed data errors on your ability to make inferences and discriminate between the three models. Where do the models match the data to within error and where do they not?
A discussion has been added on this point in section 4.2: "Considering the possible errors on observed HWMs elevation (see section 3.1), simulation errors up to 50 cm may be considered as non significant. However, these error sources are common to the three mapping approaches and should not result in any differences in the results obtained with the three methods. "

15. Line 334. Was there any supercritical flow at the bridge blockages?
It is likely that supercritical flow occured due to the section reduction under the bridge, but probably on a limited section length (longitudinal slopes are not elevated in this area).

16. Line 373. Here you might want to return to the point about the push to finer resolution becoming spurious at particular scales given other uncertainties. See the excellent paper by my co-reviewer Francesco Dottori that first identified this issue:Dottori, F., Di Baldassarre, G., and Todini, E. (2013), Detailed data is welcome, but with a pinch of salt: Accuracy, precision, and uncertainty in flood inundation modeling, Water Resour. Res., 49, 6079– 6085, doi:10.1002/wrcr.20406.
We mentioned again this issue here: "However, it should be verified that the gains related to input data accuracy are not masked by other sources of uncertainty (Dottori et al., 2013)."

17. Line 378. One solution to the Near Real Time prediction problem with 2D models is to pre-compute a library of inundation simulations at different flow rates which can then be sampled extremely quickly based on the observed or forecast discharge. The advantage of this approach is that one can sample a range of flows of weight them according to their likelihood, thereby also accounting either for the uncertainty in the observed data or in the forecast. See the following paper by David Leedal for some ideas on how to do this: Leedal, D., Neal, J., Beven, K., Younger, P. and Bates, P. (2010). Visualisation approaches for communicating real-time flood forecasting level and inundation information. Journal of Flood Risk Management, 3 (2), 140-150. (10.1111/j.1753-318X.2010.01063.x).

The discussion on this important issue of real time forecasts has been developed in the following way: " Finally, the methods presented here should be of great help to provide realistic inundation scenarios and develop information about possible flash-flood impacts as a support of flood risk management policies (Merz et al., 2020; Ritter et al., 2020). However, further work is still needed to integrate these methods into real-time forecasting chains and assess their performance in this context. The errors on discharge forecasts may indeed be dominating the other sources of uncertainties, and the computation times may also be another important limiting factor. Depending on the considered inundation mapping methods, real time computations may be feasible and may improve the representation of flood-wave volumes and flood dynamics at confluences, whereas off-line libraries of inundation scenarios can be generated and sampled in real time (Dottori et al., 2017), which may help representing discharge uncertainties by selecting multiple scenarios (Leedal et al., 2010). The definition of the best real-time computation strategy is even more complex in the case of flash-floods, because of their very fast evolution dynamics. The delay necessary to run and provide forecasts may indeed highly limit the capacity of emergency services to analyse forecasts and adapt their response strategies by reference to inundation scenarios they are prepared for. Finally, an optimal compromise has probably to be found in the case of flash floods between the accuracy of inundation forecasts and the rapidity of forecast delivery."

I hope these comments are useful and I very much look forward to seeing the paper in print.
Paul Bates
University of Bristol